# Parameter-free Regret in High Probability with Heavy Tails

**Jiujia Zhang**
Electrical and Computer Engineering
Boston University
jiujiaz@bu.edu

**Ashok Cutkosky**
Electrical and Computer Engineering
Boston University
ashok@cutkosky.com

## Abstract

We present new algorithms for online convex optimization over unbounded domains that obtain parameter-free regret in high-probability given access only to potentially heavy-tailed subgradient estimates. Previous work in unbounded domains considers only in-expectation results for sub-exponential subgradients. Unlike in the bounded domain case, we cannot rely on straight-forward martingale concentration due to exponentially large iterates produced by the algorithm. We develop new regularization techniques to overcome these problems. Overall, with probability at most $\delta$, for all comparators $\mathbf{u}$ our algorithm achieves regret $\tilde{O}(\|\mathbf{u}\| T^{1/\mathfrak{p}} \log(1/\delta))$ for subgradients with bounded $\mathfrak{p}^{th}$ moments for some $\mathfrak{p} \in (1, 2]$.

## 1 Introduction

In this paper, we consider the problem of online learning with convex losses, also called online convex optimization, with heavy-tailed stochastic subgradients. In the classical online convex optimization setting, given a convex set $\mathcal{W}$, a learning algorithm must repeatedly output a vector $\mathbf{w}_t \in \mathcal{W}$, and then observe a convex loss function $\ell_t : \mathcal{W} \to \mathbb{R}$ and incur a loss of $\ell_t(\mathbf{w}_t)$. After $T$ such rounds, the algorithm's quality is measured by the *regret* with respect to a fixed competitor $\mathbf{u} \in \mathcal{W}$:

$$R_T(\mathbf{u}) = \sum_{t=1}^{T} \ell_t(\mathbf{w}_t) - \sum_{t=1}^{T} \ell_t(\mathbf{u})$$

Online convex optimization is widely applicable, and has been used to design popular stochastic optimization algorithms ([Duchi et al., 2010a, Kingma and Ba, 2014, Reddi et al., 2018]), for control of linear dynamical systems [Agarwal et al., 2019], or even building concentration inequalities [Vovk, 2007, Waudby-Smith and Ramdas, 2020, Orabona and Jun, 2021].

A popular approach to this problem reduces it to *online linear optimization* (OLO): if $\mathbf{g}_t$ is a subgradient of $\ell_t$ at $\mathbf{w}_t$, then $R_T(\mathbf{u}) \leq \sum_{t=1}^{T} \langle \mathbf{g}_t, \mathbf{w}_t - \mathbf{u} \rangle$ so that it suffices to design an algorithm that considers only linear losses $\mathbf{w} \mapsto \langle \mathbf{g}_t, \mathbf{w} \rangle$. Then, by assuming that the domain $\mathcal{W}$ has some finite diameter $D$, standard arguments show that online gradient descent [Zinkevich, 2003] and its variants achieve $R_T(\mathbf{u}) \leq O(D\sqrt{T})$ for all $\mathbf{u} \in \mathcal{W}$. See the excellent books Cesa-Bianchi and Lugosi [2006], Shalev-Shwartz [2011], Hazan [2019], Orabona [2019] for more detail.

Deviating from the classical setting, we study the more difficult case in which, (1) the domain $\mathcal{W}$ may have *infinite* diameter (such as $\mathcal{W} = \mathbb{R}^d$), and (2) instead of observing the loss $\ell_t$, the algorithm is presented only with a potentially heavy-tailed stochastic subgradient estimate $\mathbf{g}_t$ with $\mathbb{E}[\mathbf{g}_t | \mathbf{w}_t] \in \partial \ell_t(\mathbf{w}_t)$. Our goal is to develop algorithms that, with high probability, obtain essentially the same regret bound that would be achievable even if the full information was available.

Considering only the setting of infinite diameter $\mathcal{W}$ with *exact* subgradients $\mathbf{g}_t \in \partial \ell_t(\mathbf{w}_t)$, past work has achieved bounds of the form $R_T(\mathbf{u}) \leq \tilde{O}(\epsilon + \|\mathbf{u}\|\sqrt{T})$ for all $\mathbf{u} \in \mathcal{W}$ simultaneously for any

36th Conference on Neural Information Processing Systems (NeurIPS 2022).

user-specified $\epsilon$, directly generalizing the $O(D\sqrt{T})$ rate available when $D < \infty$ [Orabona and Pál, 2016, Cutkosky and Orabona, 2018, Foster et al., 2017, Mhammedi and Koolen, 2020, Chen et al., 2021]. As such algorithms do not require knowledge of the norm $\|\mathbf{u}\|$ that is usually used to specify a learning rate for gradient descent, we will call them *parameter-free*. Note that such algorithms typically guarantee constant $R_T(0)$, which is not achieved by any known form of gradient descent.

While parameter-free algorithms appear to fully generalize the finite-diameter case, they fall short when $\mathbf{g}_t$ is a stochastic subgradient estimate. In particular, lower-bounds suggest that parameter-free algorithms must require Lipschitz $\ell_t$ [Cutkosky and Boahen, 2017], which means that care must be taken when using $\mathbf{g}_t$ with unbounded noise as this may make $\ell_t$ "appear" to be non-Lipschitz. In the case of *sub-exponential* $\mathbf{g}_t$, Jun and Orabona [2019], van der Hoeven [2019] provide parameter-free algorithms that achieve $\mathbb{E}[R_T(\mathbf{u})] \leq \tilde{O}(\epsilon + \|\mathbf{u}\|\sqrt{T})$, but these techniques do not easily extend to heavy-tailed $\mathbf{g}_t$ or to high-probability bounds. The high-probability statement is particularly elusive (even with sub-exponential $\mathbf{g}_t$) because standard martingale concentration approaches appear to fail spectacularly. This failure may be counterintuitive: for *finite diameter* $\mathcal{W}$, one can observe that $\langle \mathbf{g}_t - \mathbb{E}[\mathbf{g}_t], \mathbf{w}_t - \mathbf{u} \rangle$ forms a martingale difference sequence with variance determined by $\|\mathbf{w}_t - \mathbf{u}\| \leq D$, which allows for relatively straightforward high-probability bounds. However, parameter-free algorithms typically exhibit *exponentially growing* $\|\mathbf{w}_t\|$ in order to compete with all possible scales of $\|\mathbf{u}\|$, which appears to stymie such arguments.

Our work overcomes these issues. Requiring only that $\mathbf{g}_t$ have a bounded $\mathfrak{p}^{th}$ moment for some $\mathfrak{p} \in (1, 2]$, we devise a new algorithm whose regret with probability at least $1 - \delta$ is $R_T(\mathbf{u}) \leq \tilde{O}(\epsilon + \|\mathbf{u}\|T^{1/\mathfrak{p}}\log(1/\delta))$ for all $\mathbf{u}$ simultaneously. The $T^{1/\mathfrak{p}}$ dependency is unimprovable [Bubeck et al., 2013, Vural et al., 2022]. Moreover, we achieve these results simply by adding novel and carefully designed regularizers to the losses $\ell_t$ in a way that converts any parameter-free algorithm with sufficiently small regret into one with the desired high probability guarantee.

**Motivation:** *High-probability* analysis is appealing since it provides a confidence guarantee for an algorithm over a single run. This is crucially important in the online setting in which we must make irrevocable decisions. It is also important in the standard stochastic optimization setting encountered throughout machine learning as it ensures that even a single potentially very expensive training run will produce a good result. (See Harvey et al. [2019], Li and Orabona [2020], Madden et al. [2020], Kavis et al. [2022] for more discussion on the importance of high-probability bounds.) This goal naturally synergizes with the overall objective of *parameter-free* algorithms, which attempt to provide the best-tuned performance after a single pass over the data. In addition, we consider the presence of *heavy-tailed* stochastic gradients, which are empirically observed in large neural network architectures Zhang et al. [2020], Zhou et al. [2020].

**Contribution and Organization:** After formally introducing and discussing our setup in Sections 2, we then proceed to conduct an initial analysis for the 1-D case $\mathcal{W} = \mathbb{R}$ in 3. First (Section 4), we introduce a parameter-free algorithm for *sub-exponential* $g_t$ that achieves regret $\tilde{O}(\epsilon + |u|\sqrt{T})$ in high probability. This already improves significantly on prior work, and is accomplished by introducing a novel regularizer that "cancels" some unbounded martingale concentration terms, a technique that may have wider application. Secondly (Section 5), we extend to *heavy-tailed* $g_t$ by employing clipping, which has been used in prior work on optimization [Bubeck et al., 2013, Gorbunov et al., 2020, Zhang et al., 2020, Cutkosky and Mehta, 2021] to convert heavy-tailed estimates into sub-exponential ones. This clipping introduces some bias that must be carefully offset by yet another novel regularization (which may again be of independent interest) in order to yield our final $\tilde{O}(\epsilon + |u|T^{1/\mathfrak{p}})$ parameter-free regret guarantee. Finally (Section 6), we extend to arbitrary dimensions via the reduction from Cutkosky and Orabona [2018].

## 2 Preliminaries

Our algorithms interact with an adversary in which for $t = 1 \ldots T$ the algorithm first outputs a vector $\mathbf{w}_t \in \mathcal{W}$ for $\mathcal{W}$ a convex subset of some real Hilbert space, and then the adversary chooses a convex and $G$-Lipschitz loss function $\ell_t : \mathcal{W} \to \mathbb{R}$ and a distribution $P_t$ such that for $\mathbf{g}_t \sim P_t$, $\mathbb{E}[\mathbf{g}_t] \in \partial\ell_t(\mathbf{w}_t)$ and $\mathbb{E}[\|\mathbf{g}_t - \mathbb{E}[\mathbf{g}_t]\|^{\mathfrak{p}}] \leq \sigma^{\mathfrak{p}}$ for some $\mathfrak{p} \in (1, 2]$. The algorithm then observes a random sample $\mathbf{g}_t \sim P_t$. After $t$ rounds, we compute the *regret*, which is a function $R_t(\mathbf{u}) = \sum_{i=1}^{t} \ell_i(\mathbf{w}_i) - \ell_i(\mathbf{u})$. Our goal is to guarantee $R_T(\mathbf{u}) \leq \epsilon + \tilde{O}(\|\mathbf{u}\|T^{1/\mathfrak{p}})$ for all $\mathbf{u}$ simultaneously with high probability.

Throughout this paper we will employ the notion of a *sub-exponential* random sequence:

**Definition 1.** *Suppose $\{X_t\}$ is a sequence of random variables adapted to a filtration $\mathcal{F}_t$ such that $\{X_t, \mathcal{F}_t\}$ is a martingale difference sequence. Further, suppose $\{\sigma_t, b_t\}$ are random variables such that $\sigma_t, b_t$ are both $\mathcal{F}_{t-1}$-measurable for all $t$. Then, $\{X_t, \mathcal{F}_t\}$ is $\{\sigma_t, b_t\}$ sub-exponential if*

$$\mathbb{E}[\exp(\lambda X_t)|\mathcal{F}_{t-1}] \leq \exp(\lambda^2 \sigma_t^2 / 2)$$

*almost everywhere for all $\mathcal{F}_{t-1}$-measurable $\lambda$ satisfying $\lambda < 1/b_t$.*

We drop the subscript $t$ when we have uniform (not time-varying) sub-exponential parameters $(\sigma, b)$. We use bold font ($\mathbf{g}_t$) to refer to vectors and normal font ($g_t$) to refer to scalars. Occasionally, we abuse notation to write $\nabla \ell_t(\mathbf{w}_t)$ for an arbitrary element of $\partial \ell_t(\mathbf{w}_t)$.

We present our results using $O(\cdot)$ to hide constant factors, and $\tilde{O}(\cdot)$ to hide log factors (such as some power of $\log T$ dependence) in the main text, the exact results are left at the last line of the proof for interested readers.

Finally, observe that by the unconstrained-to-constrained conversion of Cutkosky and Orabona [2018], we need only consider the case that $\mathcal{W}$ is an entire vector space. By solving the problem for this case, the reduction implies a high-probability regret algorithm for any convex $\mathcal{W}$.

## 3  Challenges

A reader experienced with high probability bounds in online optimization may suspect that one could apply fairly standard approaches such as gradient clipping and martingale concentration to easily achieve high probability bounds with heavy tails. While such techniques do appear in our development, the story is far from straightforward. In this section, we will outline these non-intuitive difficulties. For a further discussion, see Section 3 of Jun and Orabona [2019].

For simplicity, consider $w_t \in \mathbb{R}$. Before attempting a high probability bound, one may try to derive a regret bound in expectation with heavy-tailed (or even light-tailed) gradient $g_t$ via the following calculation:

$$\mathbb{E}[R_T(u)] = \mathbb{E}\left[\sum_{t=1}^{T} \ell_t(w_t) - \ell_t(u)\right] \leq \sum_{t=1}^{T} \mathbb{E}\left[\langle g_t, w_t - u\rangle\right] + \sum_{t=1}^{T} \mathbb{E}\left[\langle \nabla \ell_t(w_t) - g_t, w_t - u\rangle\right]$$

The second sum from above vanishes, so one is tempted to send $g_t$ directly to some existing parameter-free algorithm to obtain low regret. Unfortunately, most parameter-free algorithms require a uniform bound on $|g_t|$ - even a *single* bound-violating $g_t$ could be catastrophic [Cutkosky and Boahen, 2017]. With heavy-tailed $g_t$, we are quite likely to encounter such a bound-violating $g_t$ for any reasonable uniform bound. In fact, the issue is difficult even for light-tailed $g_t$, as described in detail by Jun and Orabona [2019].

A natural approach to overcome this uniform bound issue is to incorporate some form of clipping, a commonly used technique controlling for heavy-tailed subgradients. The clipped subgradient $\hat{g}_t$ is defined below with a positive clipping parameter $\tau$ as:

$$\hat{g}_t = \frac{g_t}{|g_t|} \min(\tau, |g_t|)$$

If we run algorithms on uniformly bounded $\hat{g}_t$ instead, the expected regret can now be written as:

$$\mathbb{E}[R_T(u)] \leq \underbrace{\sum_{t=1}^{T} \mathbb{E}\left[\langle \hat{g}_t, w_t - u\rangle\right]}_{\text{parameter-free regret}} + \underbrace{\sum_{t=1}^{T} \mathbb{E}\left[\langle \mathbb{E}[\hat{g}_t] - \hat{g}_t, w_t - u\rangle\right]}_{\text{martingale concentration?}} + \underbrace{\sum_{t=1}^{T} \mathbb{E}\left[\langle \nabla \ell_t(w_t) - \mathbb{E}[\hat{g}_t], w_t - u\rangle\right]}_{\text{bias}}$$

(1)

Since $|\hat{g}_t| \leq \tau$, the first term can in fact be controlled for appropriate $\tau$ at a rate of $\tilde{O}(\epsilon + |u|\sqrt{T})$ using sufficiently advanced parameter-free algorithms (e.g. Cutkosky and Orabona [2018]). However, now bias accumulates in the last term, which is difficult to bound due to the dependency on $w_t$. On the surface, understanding this dependency appears to require detailed (and difficult) analysis of the

dynamics of the parameter-free algorithm. In fact, from naive inspection of the updates for standard parameter-free algorithms, one expects that $|w_t|$ could actually grow exponentially fast in $t$, leading to a very large bias term.

Finally, disregarding these challenges faced even in expectation, to derive a high-probability bound the natural approach is to bound the middle sum in (1) via some martingale concentration argument. Unfortunately, the variance process for this martingale depends on $w_t$ just like the bias term. In fact, this issue appears even if the original $g_t$ already have bounded norm, which is the most extreme version of *light* tails! Thus, we again appear to encounter a need for small $w_t$, which may instead grow exponentially. In summary, the unbounded nature of $w_t$ makes dealing with any kind of stochasticity in the $g_t$ very difficult. In this work we will develop techniques based on regularization that intuitively force the $w_t$ to behave well, eventually enabling our high-probability regret bounds.

## 4   Bounded Sub-exponential Noise via Cancellation

In this section, we describe how to obtain regret bound in high probability for stochastic subgradients $g_t$ for which $\mathbb{E}[g_t^2] \leq \sigma^2$ and $|g_t| \leq b$ for some $\sigma$ and $b$ (in particular, $g_t$ exhibits $(\sigma, 4b)$ sub-exponential noise). We focus on the 1-dimensional case with $\mathcal{W} = \mathbb{R}$. The extension to more general $\mathcal{W}$ is covered in Section 6. Our method involves two coordinated techniques. First, we introduce a carefully designed regularizer $\psi_t$ such that *any algorithm* that achieves low regret with respect to the losses $w \mapsto g_t w + \psi_t(w)$ will automatically ensure low regret with high probability on the original losses $\ell_t$. Unfortunately, $\psi_t$ is not Lipschitz and so it is still not obvious how to obtain low regret. We overcome this final issue by an "implicit" modification of the optimistic parameter-free algorithm of Cutkosky [2019]. Our overall goal is a regret bound of $R_T(u) \leq \tilde{O}(\epsilon + |u|(\sigma + G)\sqrt{T} + b|u|)$ for all $u$ with high probability. Note that with this bound, $b$ can be $O(\sqrt{T})$ before it becomes a significant factor in the regret.

Let us proceed to sketch the first (and most critical) part of this procedure: Define $\epsilon_t = \nabla \ell_t(w_t) - g_t$, so that $\epsilon_t$ captures the "noise" in the gradient estimate $g_t$. In this section, we assume that $\epsilon_t$ is $(\sigma, 4b)$ sub-exponential for all $t$ for some given $\sigma, b$ and $|g_t| \leq b$. Then we can write:

$$
\begin{aligned}
R_T(u) &\leq \sum_{t=1}^{T} \langle \nabla \ell_t(w_t), w_t - u \rangle = \sum_{t=1}^{T} \langle g_t, w_t - u \rangle + \sum_{t=1}^{T} \langle \epsilon_t, w_t \rangle - \sum_{t=1}^{T} \langle \epsilon_t, u \rangle \\
&\leq \sum_{t=1}^{T} \langle g_t, w_t - u \rangle + \underbrace{\left| \sum_{t=1}^{T} \epsilon_t w_t \right| + |u| \left| \sum_{t=1}^{T} \epsilon_t \right|}_{\text{"noise term", NOISE}}
\end{aligned}
\tag{2}
$$

Now, the natural strategy is to run an OLO algorithm $\mathcal{A}$ on the observed $g_t$, which will obtain some regret $R_T^{\mathcal{A}}(u) = \sum_{t=1}^{T} \langle g_t, w_t - u \rangle$, and then show that the remaining NOISE terms are small. To this end, from sub-exponential martingale concentration, we might hope to show that with probability $1 - \delta$, we have an identity similar to:

$$
\text{NOISE} \leq \sigma \sqrt{\sum_{t=1}^{T} w_t^2 \log(1/\delta)} + b \max_t |w_t| \log(1/\delta) + |u|\sigma \sqrt{T \log(1/\delta)} + |u| b \log(1/\delta)
$$

The dependency of $|u|$ above appears to be relatively innocuous as it only contributes $\tilde{O}(|u|\sigma\sqrt{T} + |u|b)$ to the regret. The $w_t$-dependent term is more difficult as it involves a dependency on the algorithm $\mathcal{A}$. This captures the complexity of our unbounded setting: in a *bounded domain*, the situation is far simpler as we can uniformly bound $|w_t| \leq D$, ideally leaving us with an $\tilde{O}(D\sqrt{T})$ bound overall.

Unfortunately, in the unconstrained case, $|w_t|$ could grow exponentially ($|w_t| \sim 2^t$) even when $u$ is very small, so we cannot rely on a uniform bound. In fact, even in the finite-diameter case, if we wish to guarantee $R_T(0) \leq \epsilon$, the bound $|w_t| \leq D$ is still too coarse. The resolution is to instead feed the algorithm $\mathcal{A}$ a *regularized* loss $\hat{\ell}_t(w) = \langle g_t, w \rangle + \psi_t(w)$, where $\psi_t$ will "cancel" the $w_t$ dependency in the martingale concentration. That is, we now define $R_T^{\mathcal{A}}(u) = \sum_{t=1}^{T} \hat{\ell}_t(w_t) - \hat{\ell}_t(u)$

and rearrange:

$$\sum_{t=1}^{T} \langle g_t, w_t - u \rangle \le R_T^{\mathcal{A}}(u) - \sum_{t=1}^{T} \psi_t(w_t) + \sum_{t=1}^{T} \psi_t(u) \tag{3}$$

And now combine equations (2) and (3):

$$R_T(u) \le R_T^{\mathcal{A}}(u) - \sum_{t=1}^{T} \psi_t(w_t) + \sum_{t=1}^{T} \psi_t(u) + \text{NOISE}$$

$$\le R_T^{\mathcal{A}}(u) + \sigma \sqrt{\sum_{t=1}^{T} w_t^2 \log(1/\delta)} + b \max_t |w_t| \log(1/\delta) - \sum_{t=1}^{T} \psi_t(w_t)$$

$$+ |u|\sigma\sqrt{T \log(1/\delta)} + |u| b \log(1/\delta) + \sum_{t=1}^{T} \psi_t(u) \tag{4}$$

From this, we can read off the desired properties of $\psi_t$: (1) $\psi_t$ should be large enough that $\sum_{t=1}^{T} \psi_t(w_t) \ge \sigma\sqrt{\sum_{t=1}^{T} w_t^2 \log(1/\delta)} + b \max_t |w_t| \log(1/\delta)$, (2) $\psi_t$ should be small enough that $\sum_{t=1}^{T} \psi_t(u) \le \tilde{O}(|u|\sqrt{T})$, and (3) $\psi_t$ should be such that $R_T^{\mathcal{A}}(u) = \tilde{O}(\epsilon + |u|\sqrt{T})$ for an appropriate algorithm $\mathcal{A}$. If we can exhibit a $\psi_t$ satisfying all three properties, we will have developed a regret bound of $\tilde{O}(\epsilon + |u|\sqrt{T})$ in high probability.

It turns out that the modified Huber loss $r_t(w)$ defined in equation (5) and (6) with appropriately chosen constants $c_1, c_2, p_1, p_2, \alpha_1, \alpha_2$ satisfies criterion (1) and (2).

$$r_t(w; c, p, \alpha_0) = \begin{cases} c\left(p|w| - (p-1)|w_t|\right) \frac{|w_t|^{p-1}}{(\sum_{i=1}^{t} |w_i|^p + \alpha_0^p)^{1-1/p}}, & |w| > |w_t| \\ c|w|^p \frac{1}{(\sum_{i=1}^{t} |w_i|^p + \alpha_0^p)^{1-1/p}}, & |w| \le |w_t| \end{cases} \tag{5}$$

$$\psi_t(w) = r_t(w; c_1, p_1, \alpha_1) + r_t(w; c_2, p_2, \alpha_2) \tag{6}$$

Let us take a moment to gain some intuition for these functions $r_t$ and $\psi_t$. First, observe that $r_t$ is always continuously differentiable, and that $r_t$'s definition requires knowledge of $w_t$. This is acceptable because online learning algorithms must be able to handle even adaptively chosen losses. In particular, consider the $p = 2$ case, $r_t(w; c, 2, \alpha)$ for some positive constants $c$ and $\alpha$. We plot this function in Figure 1, where one can see that $r_t$ grows quadratically for $|w| \le |w_t|$, but grows only linearly afterwards so that $r_t$ is Lipschitz.

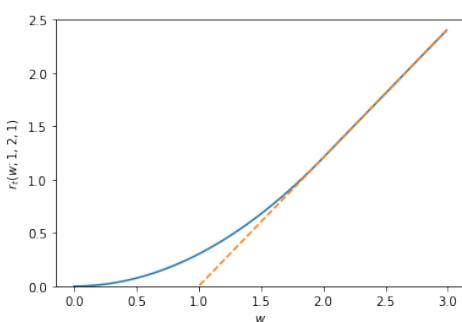

Eventually, in Lemma 13 we will show that this functions satisfies

$$\sum_{t=1}^{T} r_t(w_t; c, 2, \alpha) \ge c\sqrt{\sum_{t=1}^{T} w_t^2} - \alpha$$

$$\sum_{t=1}^{T} r_t(u : c, 2, \alpha) \le \tilde{O}(u\sqrt{T})$$

Figure 1: $r_t(w; 1, 2, 1)$ when $\sum_{i=1}^{t} w_i^2 = 10$ and $w_t = 2$. The dashed line has slope $cp \frac{|w_t|^{p-1}}{\left(\sum_{i=1}^{t} |w_i|^p + \alpha_0^p\right)^{1/p}}$, so that $r_t$ is quadratic for $|w| \le |w_t|$ and linear otherwise. Notice that $w_t$ is a constant used to define $r_t$ - it is not the argument of the function.

so that for appropriate choice of $c$ and $\alpha$, $r_t(w; c, 2, \alpha)$ will cancel the $O(\sqrt{\sum_{t=1}^{T} w_t^2})$ martingale concentration term while not adding too much to the regret - it satisfies criteria (1) and (2). The lower-bound follows from the standard inequality $\sqrt{a+b} \le \sqrt{a} + \frac{b}{\sqrt{a+b}}$ since $r_t(w_t) = c \frac{w_t^2}{\sqrt{\alpha^2 + \sum_{i=1}^{t} w_i^2}}$. The upper-bound is more subtle, and involves the piece-wise definition. For simplicity, suppose it were true that either $|w_t| < |u|$ for all $t$ or $|w_t| \ge |u|$ for all $t$. In the former case,

$\sum_{t=1}^{T} r_t(u) = O\left(|u| \sum_{t=1}^{T} \frac{|w_t|}{\sqrt{\alpha + \sum_{i=1}^{t} w_i^2}}\right)$, which via algebraic manipulation can be bounded as $\tilde{O}(|u|\sqrt{T})$. In the latter case, we have $\sum_{t=1}^{T} \frac{|u|^2}{\sqrt{\alpha^2 + \sum_{i=1}^{t} w_i^2}} \leq \sum_{t=1}^{T} \frac{|u|^2}{\sqrt{\alpha^2 + tu^2}} = \tilde{O}(|u|\sqrt{T})$ so that both cases result in the desired bound on $\sum_{t=1}^{T} r_t(u)$. The general setting is handled by partitioning the sum into two sets depending on whether $|w_t| \leq |u|$. In order to cancel the $\max_t |w_t|$ term in the martingale concentration, we employ $p = \log T$. This choice is motivated by the observation that $\|\mathbf{v}\|_{\log T} \in [\|\mathbf{v}\|_\infty, \exp(1)\|\mathbf{v}\|_\infty]$ for all $\mathbf{v} \in \mathbb{R}^{\log T}$. With this identity in hand, the argument is very similar to the $p = 2$ case.

The correct values for the constants are provided in Theorem 3. Again, at a high level, the important constants are $p_1$ and $p_2$. With $p_1 = 2$, we allow $\sum_t r_t(w_t; p = 2)$ to cancel out the $\sqrt{\sum_t w_t^2}$ martingale concentration term, while with $p_2 = \log T$, $\sum_t r_t(w_t; p = \log T)$ cancels that $\max_t |w_t|$ term.

It remains to show that $\psi_t$ also allows for small $R_T^{\mathcal{A}}(u)$ and so satisfies criterion (3). Unfortunately, our setting for $c_2$ in the definition of $\psi_t$ is $\tilde{O}(b)$, which means that $\psi_t$ is $\tilde{O}(b)$-Lipschitz. Since we wish to allow for $b = \Theta(\sqrt{T})$, this means that we cannot simply let $\mathcal{A}$ linearize $\psi_t$ and apply an arbitrary OLO algorithm. Instead, we must exploit the fact that $\psi_t$ is known *before* $g_t$ is revealed. That is, algorithm $\mathcal{A}$ is chosen to exploit the structure composite loss $\hat{\ell}_t(w)$. Intuitively, the regret of a composite loss should depend only on the non-composite $g_t$ terms (as in e.g. Duchi et al. [2010b]). Our situation is slightly more complicated as $\psi_t$ depends on $w_t$ as well, but we nevertheless achieve the desired result via a modification of the parameter-free optimistic reduction in Cutkosky [2019]. For technical reasons, this algorithm still requires $|g_t| \leq b$ with probability 1, but obtains regret only $R_T^{\mathcal{A}}(u) \leq \tilde{O}(\epsilon + |u|\sigma\sqrt{T} + |u|b)$. This technical limitation is lifted in the following section.

We display the method as Algorithm 1, which provides a regularization that cancels the $|w_t|$ dependent part of the NOISE term in (7). It also allows us to control $R_T^{\mathcal{A}}(u)$ to order $\tilde{O}(\epsilon + |u|\sigma\sqrt{T} + b|u|)$ by taking account into the predictable structure of regularizer $\psi_t(w)$. The algorithm requires black-box access to two base online learning algorithms, which we denote $\mathcal{A}_1$ and $\mathcal{A}_2$ with domains $(-\infty, \infty)$ and $[0, \infty)$ respectively. These can be any algorithms that obtain so-called "second-order" parameter-free regret bounds, such as available in Cutkosky and Orabona [2018], van der Hoeven [2019], Kempka et al. [2019], Mhammedi and Koolen [2020]. Roughly speaking, the role of $\mathcal{A}_1$ is to provide an initial candidate ouput $x_t$ that is then "corrected" by $\mathcal{A}_2$ using the regularization to obtain the final $w_t$.

Following the intuition previously outlined in this section, We first provide a deterministic regret guarantee on the quantity $R_T^{\mathcal{A}}(u) = \sum_{t=1}^{T} \hat{\ell}_t(w_t) - \hat{\ell}_t(u)$ as an intermediate result (Theorem 2). Then, we provide the analysis of the full procedure of Algorithm 1 for the final high probability result (Theorem 3). Missing proofs are provided in the Appendix A and B.

---

**Algorithm 1** Sub-exponential Noisy Gradients with Optimistic Online Learning

**Require:** $E[g_t] = \nabla\ell_t(w_t), |g_t| \leq b, \mathbb{E}[g_t|w_t] \leq \sigma^2$ almost surely. Two online learning algorithms (e.g. copies of Algorithm 1 from Cutkosky and Orabona [2018]) labelled as $\mathcal{A}_1, \mathcal{A}_2$ with domains $\mathbb{R}$ and $\mathbb{R}_{\geq 0}$ respectively. Time horizon $T$, $0 < \delta \leq 1$.
1: **Initialize:**
       Constants $\{c_1, c_2, p_1, p_2, \alpha_1, \alpha_2\}$ from Theorem 3.      ▷ for defining $\psi_t$ in equation (6)
       $H = c_1 p_1 + c_2 p_2$
2: **for** $t = 1$ to $T$ **do**
3:      Receive $x'_t$ from $\mathcal{A}_1$, $y'_t$ from $\mathcal{A}_2$
4:      Rescale $x_t = x'_t/(b+H)$, $y_t = y'_t/(H(b+H))$
5:      Solve for $w_t$: $w_t = x_t - y_t\nabla\psi_t(w_t)$      ▷ The solution exists by Lemma 6
6:      Play $w_t$ to, suffer loss $\ell_t(w_t)$
7:      Receive $g_t$ with $\mathbb{E}[g_t] \in \partial\ell_t(w_t)$
8:      Compute $\psi_t(w) = r_t(w; c_1, p_1, \alpha_1) + r_t(w; c_2, p_2, \alpha_2)$ and $\nabla\psi_t(w_t)$    ▷ equations (5), (6)
9:      Send $(g_t + \nabla\psi_t(w_t))/(b+H)$ to $\mathcal{A}_1$
10:      Send $-\langle g_t + \nabla\psi_t(w_t), \nabla\psi_t(w_t)\rangle/H(b+H)$ to $\mathcal{A}_2$
11: **end for**

---

**Theorem 2.** *Suppose $\mathcal{A}_1$ ensure that given some $\epsilon > 0$ and a sequence $c_t$ with $|c_t| \le 1$:*

$$\sum_{t=1}^{T} \langle c_t, w_t - u \rangle \le \epsilon + A|u|\sqrt{\sum_{t=1}^{T} |c_t|^2 \left(1 + \log\left(\frac{|u|^2 T^C}{\epsilon^2} + 1\right)\right)} + B|u|\log\left(\frac{|u|T^C}{\epsilon} + 1\right)$$

*for all $u$ for some positive constants $A, B, C$, and that $\mathcal{A}_2$ obtains the same guarantee for all $u \ge 0$, then for $|g_t| \le b$, $|\nabla\psi_t(w_t)| \le H$, we have the following guarantee from Algorithm 1.*

$$R_T^{\mathcal{A}}(u) \le O\left[\epsilon + |u|\left(\sqrt{\max\left(0, \sum_{t=1}^{T} |g_t|^2 - |\nabla\psi_t(w_t)|^2\right)} + (b + H)\log T\right)\right]$$

Although this Theorem 2 is rather technical, the overall message is not too complicated. If we ignore the negative $|\nabla\psi_t(w_t)|^2$ terms, the bound simply says that the regret on the "composite" loss $\langle g_t, w \rangle + \psi_t(w)$ only increases with the apriori-unknown $g_t$, and *not* with $\nabla\psi_t(w_t)$. With this result, we can formalize the intuition in this section to provide the following high probability regret bound:

**Theorem 3.** *Suppose $\{g_t\}$ are stochastic subgradients such that $\mathbb{E}[g_t] \in \partial\ell_t(w_t)$, $|g_t| \le b$ and $\mathbb{E}[g_t^2|w_t] \le \sigma^2$ almost surely for all $t$. Set the following constants for $\psi_t(w)$ shown in equation (6) for any $0 < \delta \le 1$, $\epsilon > 0$,*

$$c_1 = 2\sigma\sqrt{\log\left(\frac{32}{\delta}\left[\log\left(2^{T+1}\right) + 2\right]^2\right)}, \quad c_2 = 32b\log\left(\frac{224}{\delta}\left[\log\left(1 + \frac{b}{\sigma}2^{T+2}\right) + 2\right]^2\right),$$

$$p_1 = 2, \qquad p_2 = \log T, \qquad \alpha_1 = \epsilon/c_1, \qquad \alpha_2 = \epsilon\sigma/(4b(b + H))$$

*where $H = c_1 p_1 + c_2 p_2$, $|\nabla\psi_t(w_t)| \le H$. Then, with probability at least $1 - \delta$, algorithm 1 guarantees*

$$R_T(u) \le \tilde{O}\left[\epsilon\log\frac{1}{\delta} + |u|b\log\frac{1}{\delta} + |u|\sigma\sqrt{T\log\frac{1}{\delta}}\right]$$

Note that this result is *already* of interest: prior work on parameter-free algorithms with sub-exponential noise only achieve in-expectation rather than high probability results. Of course, there is a caveat: our bound requires that $|g_t|$ be uniformly bounded by $b$. Even though $b$ could be as large as $\sqrt{T}$, this is still a mild restriction. In the next section, we remove both this restriction as well as the light tail assumption all together.

## 5 Heavy tails via Truncation

In this section, we aim to give a high probability bound for heavy-tailed stochastic gradients $\mathbf{g}_t$. Our approach builds on Section 4 by incorporating gradient clipping with a clipping parameter $\tau \in \mathbb{R}^+$.

$$\hat{\mathbf{g}}_t = \frac{\mathbf{g}_t}{\|\mathbf{g}_t\|}\min(\tau, \|\mathbf{g}_t\|)$$

We continue to consider a 1-dimensional problem in this section, replacing the norm $\|\cdot\|$ with absolute value $|\cdot|$ and $\mathbf{g}_t$ with $g_t$. The key insight is that the clipped $\hat{g}_t$ satisfies $\mathbb{E}[\hat{g}_t^2] \le 2^{\mathfrak{p}-1}\tau^{2-\mathfrak{p}}(\sigma^{\mathfrak{p}} + G^{\mathfrak{p}})$ and of course $|\hat{g}_t| \le \tau$. Hence, a high probability bound could be obtained by feeding $\hat{g}_t$ into Algorithm 1 from Section 4. Let us formally quantify the effect of this clipping:

$$R_T(u) \le \sum_{t=1}^{T} \langle \nabla\ell_t(w_t), w_t - u \rangle = \underbrace{\sum_{t=1}^{T} \langle \nabla\ell_t(w_t) - \mathbb{E}[\hat{g}_t], w_t - u \rangle}_{\text{bias}} + \underbrace{\sum_{t=1}^{T} \langle \mathbb{E}[\hat{g}_t], w_t - u \rangle}_{\text{Section 4}} \quad (7)$$

Without clipping, we would have $\mathbb{E}[\hat{g}_t] = \nabla\ell_t(w_t)$, and so if we were satisfied with an in-expectation result, the first sum above would vanish. However, with clipping, the first sum actually represents some "bias" that must be controlled even to obtain an in-expectation result, let alone high probability. We control this bias using a cancellation-by-regularization strategy analogous at a high level to the one developed in Section 4, although technically quite distinct. After dealing with the bias, we must

handle the second sum. Fortunately, since $\hat{g}_t$ is sub-exponential, bounding the second sum in high probability is precisely the problem solved in Section 4. We introduce the analysis in two elementary steps. For the purpose of bias cancellation, we define a linearized loss $\tilde{\ell}_t(w)$ with regularization function $\phi(w)$

$$\tilde{\ell}_t(w) = \langle \mathbb{E}[\hat{g}_t], w \rangle + \phi(w), \qquad \phi(w) = 2^{\mathfrak{p}-1}(\sigma^{\mathfrak{p}} + G^{\mathfrak{p}})|w|/\tau^{\mathfrak{p}-1} \qquad (8)$$

the regret in equation (7) can be re-written as

$$= \underbrace{\sum_{t=1}^{T} \left( \langle \nabla \ell_t(w_t) - \mathbb{E}[\hat{g}_t], w_t - u \rangle - \phi(w_t) + \phi(u) \right)}_{\text{bias cancellation}} + \underbrace{\sum_{t=1}^{T} \tilde{\ell}_t(w_t) - \tilde{\ell}_t(u)}_{\text{Section 4}} \qquad (9)$$

We will be able to show that the $w_t$-dependent terms of the first summation sum to a negative number and so can be dropped. This leaves only the $u$-dependent terms, which for appropriate choice of $\tau$ will be $\tilde{O}(|u|T^{1/\mathfrak{p}})$.

Note that at this point, if we were satisfied with an *in expectation* bound for heavy-tailed subgradient estimates (which would already be an interesting new result), we would not require the techniques of Section 4: we could instead define $\hat{\ell}_t(w) = \langle \hat{g}_t, w \rangle + \psi(w)$, so that the last sum is equal to $\sum_{t=1}^{T} \hat{\ell}_t(w_t) - \hat{\ell}_t(u)$ in expectation. Then, since $|\hat{\ell}_t(w_t)| \leq O(\tau)$ with probability 1, we can control $\sum_{t=1}^{T} \hat{\ell}_t(w_t) - \hat{\ell}_t(u)$ using a parameter-free algorithm obtaining regret $\tilde{O}(|u|\sqrt{\sum_{t=1}^{T} |\nabla \hat{\ell}_t(w_t)|^2} + \tau|u|)$ to bound the total expected regret, yielding a simple way to recover prior work on expected regret with sub-exponential subgradients (up to logs), while extending the results to heavy-tailed subgradients.

However, since we *do* aim for a high probability bound, we need to be more careful with the second summation. Fortunately, given that $\hat{g}_t$ is sub-exponential and bounded, and $\nabla\phi(w_t)$ is deterministic, we can supply $\hat{g}_t + \nabla\phi(w_t)$ to Algorithm 1 and then bound the sum in high probability by Theorem 3. We formalize the procedure as Algorithm 2, and its guarantee is stated in Theorem 4. The exact regret guarantee (including constants) can be found in Appendix C.

---

**Algorithm 2** Gradient clipping for $(\sigma, G)-$Heavy tailed gradients

---

**Require:** $\mathbb{E}[g_t] = \nabla\ell_t(w_t)$, $|\mathbb{E}[g_t]| \leq G$, $\mathbb{E}[|g_t - \mathbb{E}[g_t]|^{\mathfrak{p}}] \leq \sigma^{\mathfrak{p}}$ for some $\mathfrak{p} \in (1, 2]$, Time horizon $T$, gradient clipping parameter $\tau$.
1: Initialize Algorithm 1 using the parameters of Theorem 3.
2: **for** $t = 1$ to $T$ **do**
3:      Receive $w_t$ from Algorithm 1.
4:      Suffer loss $\ell_t(w_t)$, receive $g_t$
5:      Truncate $\hat{g}_t = \frac{g_t}{|g_t|}\min(\tau, |g_t|)$.
6:      Compute $\tilde{g}_t = \hat{g}_t + \nabla\phi_t(w_t)$          $\triangleright$ $\phi(w)$ is defined in (8), $\mathbb{E}[\tilde{g}_t] \in \partial\tilde{\ell}_t(w_t)$.
7:      Send $\tilde{g}_t$ to Algorithm 1 as $t^{th}$ subgradient.
8: **end for**

---

**Theorem 4.** *Suppose $\{g_t\}$ are heavy-tailed stochastic gradient such that $\mathbb{E}[g_t] \in \partial\ell_t(w_t)$, $|\mathbb{E}[g_t]| \leq G$, $\mathbb{E}[|g_t - \mathbb{E}[g_t]|^{\mathfrak{p}}] \leq \sigma^{\mathfrak{p}}$ for some $\mathfrak{p} \in (1, 2]$. If we set $\tau = T^{1/\mathfrak{p}}(\sigma^{\mathfrak{p}} + G^{\mathfrak{p}})^{1/\mathfrak{p}}$ then with probability at least $1 - \delta$, Algorithm 2 guarantees:*

$$R_T(u) \leq \tilde{O}\left[ \epsilon \log\frac{1}{\delta} + |u|T^{1/\mathfrak{p}}(\sigma + G)\log\frac{T}{\delta}\log\frac{|u|T}{\epsilon} \right]$$

Theorem 4 suggests regret with heavy-tailed gradients $g_t$ has a $\mathfrak{p}$ dependence of $\tilde{O}(T^{1/\mathfrak{p}})$, which is optimal [Bubeck et al., 2013, Vural et al., 2022].

## 6 Dimension-free Extension

So far, we have only considered 1-dimensional problems. In this section, we demonstrate the extension to dimension-free, which is achieved by using a reduction from Cutkosky and Orabona [2018]. The

original reduction extends a 1-dimensional algorithm to a dimension-free one by dissecting the problem into a "magnitude" and a "direction" learner. The direction learner is a constrained OLO algorithm $\mathcal{A}^{nd}$ which outputs a vector $\mathbf{v}_t$ with $\|\mathbf{v}_t\| \leq 1$ in response to $\mathbf{g}_1, \ldots, \mathbf{g}_{t-1}$, while the magnitude learner is an unconstrained OLO algorithm $\mathcal{A}^{1d}$ which outputs $x_t \in \mathbb{R}$ in response to $\langle \mathbf{g}_1, \mathbf{v}_1 \rangle, \ldots \langle \mathbf{g}_{t-1}, \mathbf{v}_{t-1} \rangle$. The output of the entire algorithm is $\mathbf{w}_t = x_t \mathbf{v}_t$. Suppose $\mathcal{A}_{1d}$ and $\mathcal{A}_{nd}$ have regret guarantee of $R_T^{1d}(u)$ and $R_T^{nd}(\mathbf{u})$, respectively. Then regret of the dimension-free reduction is bounded by $R_T(\mathbf{u}) \leq \|\mathbf{u}\| R_T^{nd}(\mathbf{u}/\|\mathbf{u}\|) + R_T^{1d}(\|\mathbf{u}\|)$. Thus, in order to apply this reduction we need to exhibit a $\mathcal{A}^{1d}$ and $\mathcal{A}^{nd}$ that achieves low regret on heavy-tailed losses. For the magnitude learner $\mathcal{A}^{1d}$, we use can use the 1d Algorithm 2 that we just developed. The remaining question is how to develop a direction learner that can handle heavy-tailed subgradients. Fortunately, this is much easier since the direction learner is constrained to the unit ball.

To build this direction learner, we again apply subgradient clipping, and feed the clipped subgradients to the standard FTRL algorithm with quadratic regularizer (i.e. "lazy" online gradient descent). This procedure is described in Algorithm 3. Note there is no regularization implemented in Algorithm 3 although $\hat{\mathbf{g}}_t$ induces bias. This is because of $\mathcal{A}^{nd}$ runs on the unit ball, careful tuning of $\tau$ is sufficient to control the bias - a concrete demonstration of how much more intricate the unconstrained case is! Finally, the full dimension-free reduction is displayed in Algorithm 4 with its high probability guarantee stated in Theorem 5. The details are presented in Appendix D.

---

**Algorithm 3** Unit Ball Gradient clipping with FTRL

---

**Require:** time horizon $T$, gradient clipping parameter $\tau$, regularizer weight $\eta$
1: Set $\eta = 1/\tau$
2: **for** $t = 1$ to $T$ **do**
3:      Compute $\mathbf{v}_t \in \operatorname{argmin}_{\mathbf{v}:\|\mathbf{v}\|\leq 1} \sum_{i=1}^{t-1} \langle \hat{\mathbf{g}}_t, \mathbf{v} \rangle + \frac{1}{2\eta}\|\mathbf{v}\|^2$
4:      Output $\mathbf{v}_t$, receive gradient $\mathbf{g}_t$
5:      Set $\hat{\mathbf{g}}_t = \frac{\mathbf{g}_t}{\|\mathbf{g}_t\|} \min(\tau, \|\mathbf{g}_t\|)$
6: **end for**

---

**Algorithm 4** Dimension-free Gradient clipping for $(\sigma, G)$ Heavy-tailed gradients

---

**Require:** Subgradients $\mathfrak{p}^{th}$ moment bound $\sigma^{\mathfrak{p}}$, time horizon $T$, Set Algorithm 2, 3 as $\mathcal{A}^{1d}, \mathcal{A}^{nd}$.
1: Set $\sigma_{1d} = (\sigma^{\mathfrak{p}} + 2G^{\mathfrak{p}})^{1/\mathfrak{p}}$ and $\tau_{1d} = T^{1/\mathfrak{p}}(\sigma_{1d}^{\mathfrak{p}} + G^{\mathfrak{p}})^{1/\mathfrak{p}} = T^{1/\mathfrak{p}}(\sigma^{\mathfrak{p}} + 3G^{\mathfrak{p}})^{1/\mathfrak{p}}$
2: Initialize $\mathcal{A}^{1d}$ with parameters $\sigma \leftarrow \sigma_{1d}$ and $\tau \leftarrow \tau_{1d}$
3: Initialize $\mathcal{A}^{nd}$ with parameters $\sigma \leftarrow \sigma$ and $\tau \leftarrow T^{1/\mathfrak{p}}(\sigma^{\mathfrak{p}} + G^{\mathfrak{p}})^{1/\mathfrak{p}}$.
4: **for** $t = 1$ to $T$ **do**
5:      Receive $x_t \in \mathbb{R}$ from $\mathcal{A}^{1d}$,
6:      Receive $\mathbf{v}_t \in \mathbb{R}^d, \|\mathbf{v}_t\| \leq 1$ from $\mathcal{A}^{nd}$
7:      Play output $\mathbf{w}_t = x_t \mathbf{v}_t$
8:      Suffer loss $\ell_t(\mathbf{w}_t)$, receive gradients $\mathbf{g}_t$
9:      Send $g_t = \langle \mathbf{g}_t, \mathbf{v}_t \rangle$ as the $t^{th}$ gradient to $\mathcal{A}^{1d}$
10:      Send $\mathbf{g}_t$ as the $t^{th}$ gradient to $\mathcal{A}^{nd}$
11: **end for**

---

**Theorem 5.** *Suppose that for all $t$, $\{\mathbf{g}_t\}$ are heavy-tailed stochastic subgradients satisfying $\mathbb{E}[\mathbf{g}_t] \in \partial \ell_t(\mathbf{w}_t)$, $\|\mathbb{E}[\mathbf{g}_t]\| \leq G$ and $\mathbb{E}[\|\mathbf{g}_t - \mathbb{E}[\mathbf{g}_t]\|^{\mathfrak{p}}] \leq \sigma^{\mathfrak{p}}$ for some $\mathfrak{p} \in (1, 2]$. Then, with probability at least $1 - \delta$, Algorithm 4 guarantees*

$$R_T(\mathbf{u}) = \sum_{t=1}^{T} \ell_t(\mathbf{w}_t) - \ell_t(\mathbf{u}) \leq \tilde{O}\left[\epsilon \log \frac{1}{\delta} + \|\mathbf{u}\| T^{1/\mathfrak{p}}(\sigma + G) \log \frac{T}{\delta} \log \frac{\|\mathbf{u}\|T}{\epsilon}\right]$$

**Complexity Analysis**: Algorithm 4 requires $O(d)$ space. It also requires $O(d)$ time for all operations except solving the fixed-point equation in Algorithm 1 (line 5). This can be solved via binary search to arbitrary precision $\epsilon_0$ for an overall complexity of $O(d + \log(1/\epsilon_0))$. This is essentially $O(d)$ in practice for any $d > 64$.

# 7 Conclusion

We have presented a framework for building parameter-free algorithms that achieve high probability regret bounds for heavy-tailed subgradient estimates. This improves upon prior work in several ways: high probability bounds were previously unavailable even for the restricted setting of *bounded* subgradient estimates, while even in-expectation bounds were previously unavailable for heavy-tailed subgradients. Our development required two new techniques: first, we described a regularization scheme that effectively "cancels" potentially problematic iterate-dependent variance terms arising in standard martingale concentration arguments. This allows for high probability bounds with bounded sub-exponential estimates, and we hope may be of use in other scenarios where the iterates appear in variance calculations. The second combines clipping with another new regularization scheme that "cancels" another problematic iterate-dependent *bias* term. On its own, this technique actually can be used to recover in-expectation bounds for heavy-tailed estimates.

**Limitations:** Our algorithm has several limitations that suggest open questions: first, our two regularization schemes each introduce potentially suboptimal logarithmic factors. The first one introduces a higher logarithmic dependence on $T$, while the second introduces a higher logarithmic dependence on $\|\mathbf{u}\|$ because the optimal clipping parameter $\tau$ depends on $\log(\|\mathbf{u}\|)$. Beyond this, our algorithms require knowledge of the parameters $\sigma$ and $\tau$. Adapting to an unknown value of even one of these parameters remains a challenging problem.

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
