## A Optimistic Online Learning for Predictable Regularizer

Algorithm 1 provides output $w_t$ by solving $w_t = x_t - y_t \nabla \psi_t(w_t)$, where $x_t \in \mathbb{R}, y_t \geq 0$ are output from sub-algorithms $\mathcal{A}_1$ and $\mathcal{A}_2$, $\psi_t(w)$ is defined in equation (6). Under the constants for $\psi_t(w)$ defined in Theorem 3, the following Lemma shows the existence of solution.

**Lemma 6** (Existence of Solution). *for $x_t \in \mathbb{R}, y_t \geq 0$,*

$$w_t = x_t - y_t \nabla \psi_t(w_t)$$

*where*

$$\nabla \psi_t(w) = \text{sign}(w) \sum_{j=1}^{2} k_j p_j \frac{|w|^{p_j - 1}}{(|w|^{p_j} + X_j)^{1 - 1/p_j}}$$

*for some $k_j, X_j > 0, p_j > 1$ and $j = 1, 2$. Then $w_t$ lies in the interval of $\left( x_t - y_t \sum_{j=1}^{2} k_j p_j, x_t \right]$ when $x_t \geq 0$, and in the interval of $\left[ x_t, x_t + y_t \sum_{j=1}^{2} k_j p_j \right)$. Further,*

$$h(w) = w - x_t + y_t \nabla \psi_t(w)$$

*is monotonic in $w$.*

*Proof.* We suppose that $x_t \geq 0$. The case $x_t < 0$ is entirely identical.

**case (a):** consider $y_t \neq 0$,

$$w_t = x_t - y_t \text{sign}(w_t) \sum_{j=1}^{2} k_j p_j \frac{|w_t|^{p_j - 1}}{(|w_t|^{p_j} + X_j)^{(p_j - 1)/p_j}}$$

rearrange

$$\frac{x_t - w_t}{y_t} = \text{sign}(w_t) \sum_{j=1}^{2} k_j p_j \frac{|w_t|^{p_j - 1}}{(|w_t|^{p_j} + X_j)^{(p_j - 1)/p_j}}$$

Let $f(w_t), g(w_t)$ to be the left and right handside of the last expression. Both functions are continuous in $w_t$ for under assumption of $x_t, y_t, k_j, p_j, X_j$ for $j = 1$ and 2. When $w_t^* = x_t$:

$$f(w_t^*) - g(w_t^*) = 0 - \sum_{j=1}^{2} k_j p_j \frac{|w_t^*|^{p_j - 1}}{(|w_t^*|^{p_j} + X_j)^{(p_j - 1)/p_j}} \leq 0$$

When $w_t^* = x_t - y_t \sum_{j=1}^{2} k_j p_j$:

$$f(w_t^*) - g(w_t^*) = \sum_{j=1}^{2} k_j p_j - \text{sign} \left( x_t - y_t \sum_{j=1}^{2} k_j p_j \right) \sum_{j=1}^{2} k_j p_j \frac{|w_t^*|^{p_j - 1}}{(|w_t^*|^{p_j} + X_j)^{(p_j - 1)/p_j}} > 0$$

By intermediate value Theorem $f(w_t) = g(w_t)$ at $w_t$ in between $x_t$ and $x_t - y_t \sum_{j=1}^{2} k_j p_j$.
**case (b):** when $y_t = 0$, $w_t = x_t$.

Finally, by inspection the derivative of $h(w)$ with respective to $w$ is always positive, hence is monotonic in $w$ so that we can numerically solve for $h(w_t^*) = 0$ via binary search. $\square$

Algorithm 1 requires the base algorithms $\mathcal{A}_1$ and $\mathcal{A}_2$ to satisfy a "second-order" regret bound, such as provided by Algorithm 1 of Cutkosky and Orabona [2018]. We assume the base algorithms are designed to handle only $1-$Lipschitz losses, so the following Lemma provide a simple linear transformation that allows the base algorithm to cope with any Lipschitz constant.

**Lemma 7** (Algorithm Transformation). *Suppose an algorithm $\mathcal{A}$ obtains regret $\sum_{t=1}^{T} \langle g_t, w_t - u \rangle \leq \epsilon + R_T(u)$ for some function $R_T$ for any sequence $\{g_t\}$ such that $|g_t| \leq G$. Then, given some $\bar{\epsilon} > 0$, consider the algorithm that plays $\bar{w}_t = \frac{\bar{\epsilon}G}{\epsilon\bar{G}}w_t$ in response to subgradients $\{\bar{g}_t\}$ with $|\bar{g}_t| \leq \bar{G}$, where $w_t$ is the output of $\mathcal{A}$ on the sequence $\{g_t\}$ with $g_t = \frac{G}{\bar{G}}\bar{g}_t$. This procedure ensures regret:*

$$\sum_{t=1}^{T} \langle \bar{g}_t, \bar{w}_t - u \rangle \leq \bar{\epsilon} + \frac{\bar{\epsilon}}{\epsilon} R_T \left( \frac{\epsilon\bar{G}}{\bar{\epsilon}G} u \right)$$

*Proof.* Since $|g_t| \leq G$ by construction, we have:

$$\sum_{t=1}^{T} \langle \bar{g}_t, \bar{w}_t - u \rangle = \frac{\bar{G}}{G} \sum_{t=1}^{T} \left\langle g_t, \frac{\bar{\epsilon}G}{\epsilon\bar{G}}w_t - u \right\rangle$$

$$= \frac{\bar{\epsilon}}{\epsilon} \sum_{t=1}^{T} \left\langle g_t, w_t - \frac{\epsilon\bar{G}}{\bar{\epsilon}G}u \right\rangle$$

$$\leq \bar{\epsilon} + \frac{\bar{\epsilon}}{\epsilon} R_T \left( \frac{\epsilon\bar{G}}{\bar{\epsilon}G}u \right)$$

$\square$

Intuitively, if we instantiate this Lemma with an algorithm obtaining $R_T(u) = \epsilon + |u|\sqrt{T\log(|u|T/\epsilon)} + |u|\log(|u|T/\epsilon)$ for 1-Lipschitz losses, we can obtain for any $\epsilon$, an algorithm for $G$-Lipschitz losses with regret $\epsilon + |u|G\sqrt{T\log(|u|GT/\epsilon)} + |u|G\log(|u|GT/\epsilon)$.

We are now at the stage to prove Theorem 2. We restate the Theorem for reference, followed by its proof.

**Theorem 2.** *Suppose $\mathcal{A}_1$ ensure that given some $\epsilon > 0$ and a sequence $c_t$ with $|c_t| \leq 1$:*

$$\sum_{t=1}^{T} \langle c_t, w_t - u \rangle \leq \epsilon + A|u|\sqrt{\sum_{t=1}^{T} |c_t|^2 \left( 1 + \log\left( \frac{|u|^2 T^C}{\epsilon^2} + 1 \right) \right)} + B|u|\log\left( \frac{|u|T^C}{\epsilon} + 1 \right)$$

*for all $u$ for some positive constants $A, B, C$, and that $\mathcal{A}_2$ obtains the same guarantee for all $u \geq 0$, then for $|g_t| \leq b$, $|\nabla\psi_t(w_t)| \leq H$, we have the following guarantee from Algorithm 1,*

$$R_T^{\mathcal{A}}(u) \leq O\left[ \epsilon + |u| \left( \sqrt{\max\left( 0, \sum_{t=1}^{T} |g_t|^2 - |\nabla\psi_t(w_t)|^2 \right)} + (b+H)\log T \right) \right]$$

*Proof.* The proof is similar to optimistic reduction in Cutkosky [2019], which combines regret guarantees from two online learning algorithms. First, we observe that $\mathcal{A}_1$ outputs $x'_t$ from line 3 and receives gradients $(g_t + \nabla\psi_t(w_t))/(b+H) \leq 1$ from line 9 in Algorithm 1. Hence we apply Lemma 7 by choosing $\epsilon = \bar{\epsilon}$, and set $G = 1$, $\bar{G} = b + H$, we have the following holds for any $u$,

$$R_T^1(u) = \sum_{t=1}^{T} \langle g_t + \nabla\psi_t(w_t), x_t - u \rangle$$

$$\leq \epsilon + A|u|\sqrt{\sum_{t=1}^{T} |g_t + \nabla\psi_t(w_t)|^2 \left[ 1 + \log\left( \frac{(b+H)^2|u|^2 T^C}{\epsilon^2} + 1 \right) \right]}$$

$$+ B(b+H)|u|\log\left( \frac{(b+H)|u|T^C}{\epsilon} + 1 \right)$$

Similarly for $\mathcal{A}_2$ outputs $y'_t$ and receives $\frac{-\langle g_t + \nabla\psi_t(w_t), \nabla\psi_t(w_t) \rangle}{H(b+H)}$, Hence use Lemma 7 by setting $\epsilon = \bar{\epsilon}$, $G = 1$ and $\bar{G} = H(b+H)$, we have the following for all $y_\star$:

$$R_T^2(y_\star) = \sum_{t=1}^{T} -\langle g_t + \nabla\psi_t(w_t), \nabla\psi_t(w_t) \rangle (y_t - y_\star)$$

$$\leq \epsilon + A|y_\star| \sqrt{\sum_{t=1}^{T} \langle g_t + \nabla\psi_t(w_t), \nabla\psi_t(w_t)\rangle^2 \left[1 + \log\left(\frac{(b+H)^2 H^2 |y_\star|^2 T^C}{\epsilon^2} + 1\right)\right]}$$

$$+ B(b+H)H|y_\star|\log\left(\frac{(b+H)H|y_\star|T^C}{\epsilon} + 1\right)$$

The relationship between the $R_T^A(u)$ bounded by linearized loss and $R_T^1(u)$, $R_T^2(y_\star)$ is revealed:

$$R_T^A(u) \leq \sum_{t=1}^{T} \langle g_t + \nabla\psi_t(w_t), w_t - u\rangle$$

$$= \sum_{t=1}^{T} \langle g_t + \nabla\psi_t(w_t), x_t - u\rangle - y_t\langle g_t + \nabla\psi_t(w_t), \nabla\psi_t(w_t)\rangle$$

$$\leq \inf_{y_\star \geq 0} R_T^1(u) + R_T^2(y_\star) - y_\star\sum_{t=1}^{T}\langle g_t + \nabla\psi_t(w_t), \nabla\psi_t(w_t)\rangle$$

use identity $-2\langle a, b\rangle = \|a - b\|^2 - \|a\|^2 - \|b\|^2$

$$= \inf_{y_\star \geq 0} R_T^1(u) + R_T^2(y_\star) + \frac{y_\star}{2}\sum_{t=1}^{T} |g_t|^2 - |g_t + \nabla\psi_t(w_t)|^2 - |\nabla\psi_t(w_t)|^2$$

$$\leq \inf_{y_\star \geq 0} 2\epsilon + A|u|\sqrt{\sum_{t=1}^{T}|g_t + \nabla\psi_t(w_t)|^2\left[1 + \log\left(\frac{(b+H)^2|u|^2 T^C}{\epsilon^2} + 1\right)\right]}$$

$$+ B(b+H)|u|\log\left[\frac{(b+H)|u|T^C}{\epsilon} + 1\right] + B(b+H)H|y_\star|\log\left[\frac{(b+H)H|y_\star|T^C}{\epsilon} + 1\right]$$

$$+ A|y_\star|\sqrt{\sum_{t=1}^{T}\langle g_t + \nabla\psi_t(w_t), \nabla\psi_t(w_t)\rangle^2\left[1 + \log\left(\frac{(b+H)^2 H^2 |y_\star|^2 T^C}{\epsilon^2} + 1\right)\right]}$$

$$+ \frac{y_\star}{2}\sum_{t=1}^{T} |g_t|^2 - |g_t + \nabla\psi_t(w_t)|^2 - |\nabla\psi_t(w_t)|^2$$

let $X = \sum_{t=1}^{T}|g_t + \nabla\psi_t(w_t)|^2$

$$\leq \inf_{y_\star \geq 0}\sup_{X \geq 0} 2\epsilon + A|u|\sqrt{X\left[1 + \log\left(\frac{(b+H)^2|u|^2 T^C}{\epsilon^2} + 1\right)\right]}$$

$$+ B(b+H)|u|\log\left[\frac{(b+H)|u|T^C}{\epsilon} + 1\right] + B(b+H)H|y_\star|\log\left[\frac{(b+H)H|y_\star|T^C}{\epsilon} + 1\right]$$

$$+ A|y_\star|\sqrt{XH^2\left[1 + \log\left(\frac{(b+H)^2 H^2 |y_\star|^2 T^C}{\epsilon^2} + 1\right)\right]}$$

$$+ \frac{y_\star}{2}\sum_{t=1}^{T}\left(|g_t|^2 - |\nabla\psi_t(w_t)|^2\right) - \frac{y_\star}{2}X$$

$$\leq \inf_{y_\star \geq 0}\sup_{X \geq 0}\sup_{Z \geq 0} 2\epsilon + A|u|\sqrt{X\left[1 + \log\left(\frac{(b+H)^2|u|^2 T^C}{\epsilon^2} + 1\right)\right]}$$

$$+ B(b+H)|u|\log\left[\frac{(b+H)|u|T^C}{\epsilon} + 1\right] + B(b+H)H|y_\star|\log\left[\frac{(b+H)H|y_\star|T^C}{\epsilon} + 1\right]$$

$$+ A|y_\star|\sqrt{ZH^2\left[1 + \log\left(\frac{(b+H)^2 H^2 |y_\star|^2 T^C}{\epsilon^2} + 1\right)\right]}$$

$$+ \frac{y_\star}{2} \sum_{t=1}^{T} \left( |g_t|^2 - |\nabla \psi_t(w_t)|^2 \right) - \frac{y_\star}{4}(X + Z)$$

set

$$y_\star = \min \left( \frac{2A|u|\sqrt{1 + \log((b+H)^2|u|^2 T^C/\epsilon^2 + 1)}}{\sqrt{\max(0, \sum_{t=1}^{T} \left( |g_t|^2 - |\nabla \psi_t(w_t)|^2 \right))}}, \frac{|u|}{H} \right)$$

$$\leq \sup_{X \geq 0} \sup_{Z \geq 0} 2\epsilon + A|u|\sqrt{X \left[ 1 + \log \left( \frac{(b+H)^2|u|^2 T^C}{\epsilon^2} + 1 \right) \right]}$$

$$+ B(b+H)|u| \log \left[ \frac{(b+H)|u|T^C}{\epsilon} + 1 \right] - \frac{y_\star}{4}(X + Z)$$

$$+ B(b+H)|u| \log \left[ \frac{(b+H)|u|T^C}{\epsilon} + 1 \right]$$

$$+ A|y_\star|\sqrt{ZH^2 \left[ 1 + \log \left( \frac{(b+H)^2 H^2 |y_\star|^2 T^C}{\epsilon^2} + 1 \right) \right]}$$

$$+ A|u|\sqrt{1 + \log(\frac{(b+H)^2|u|^2 T^C}{\epsilon^2} + 1)} \sqrt{\max(0, \sum_{t=1}^{T} \left( |g_t|^2 - |\nabla \psi_t(w_t)|^2 \right))}$$

For $a, b > 0$, $\sup_x a\sqrt{x} - bx = a^2/4b$, apply the identity to both $\sup_{X>0}, \sup_{Z>0}$

$$\leq 2\epsilon + A^2|u|^2 \left[ 1 + \log \left( \frac{(b+H)^2|u|^2 T^C}{\epsilon^2} + 1 \right) \right] / y_\star$$

$$+ A^2|y_\star|H^2 \left[ 1 + \log \left( \frac{(b+H)^2 H^2 |y_\star|^2 T^C}{\epsilon^2} + 1 \right) \right]$$

$$+ 2B(b+H)|u| \log \left[ \frac{(b+H)|u|T^C}{\epsilon} + 1 \right]$$

$$+ A|u|\sqrt{1 + \log(\frac{(b+H)^2|u|^2 T^C}{\epsilon^2} + 1)} \sqrt{\max(0, \sum_{t=1}^{T} \left( |g_t|^2 - |\nabla \psi_t(w_t)|^2 \right))}$$

substitute $y_\star$

$$\leq 2\epsilon + \frac{A}{2}|u|\sqrt{\left[ 1 + \log \left( \frac{(b+H)^2|u|^2 T^C}{\epsilon^2} + 1 \right) \right]} \sqrt{\max(0, \sum_{t=1}^{T} \left( |g_t|^2 - |\nabla \psi_t(w_t)|^2 \right))}$$

$$+ A^2|u|H \left[ 1 + \log \left( \frac{(b+H)^2|u|^2 T^C}{\epsilon^2} + 1 \right) \right] + 2B(b+H)|u| \log \left[ \frac{(b+H)|u|T^C}{\epsilon} + 1 \right]$$

$$+ A|u|\sqrt{1 + \log(\frac{(b+H)^2|u|^2 T^C}{\epsilon^2} + 1)} \sqrt{\max(0, \sum_{t=1}^{T} \left( |g_t|^2 - |\nabla \psi_t(w_t)|^2 \right))}$$

$$\leq 2\epsilon + \frac{3A}{2}|u|\sqrt{\left[ 1 + \log \left( \frac{(b+H)^2|u|^2 T^C}{\epsilon^2} + 1 \right) \right]} \sqrt{\max(0, \sum_{t=1}^{T} \left( |g_t|^2 - |\nabla \psi_t(w_t)|^2 \right))}$$

$$+ |u| \left( A^2 H + 2B(b+H) \right) \left[ 1 + \log \left( \frac{(b+H)^2|u|^2 T^C}{\epsilon^2} + 1 \right) \right]$$

Define a constant $N$

$$N = 1 + \log \left( \frac{(b+H)^2|u|^2 T^C}{\epsilon^2} + 1 \right)$$

Then, $R_T^{\mathcal{A}}(u)$ can be written as,

$$R_T^{\mathcal{A}}(u) \le \sum_{t=1}^{T} \langle g_t + \nabla\psi_t(w_t), w_t - u \rangle$$

$$\le 2\epsilon + |u| \left[ \frac{3A}{2} \sqrt{N \max\left(0, \sum_{t=1}^{T} |g_t|^2 - |\nabla\psi_t(w_t)|^2\right)} + \left(A^2 H + 2B(b+H)\right) N \right]$$

$\square$

The following Lemma shows the magnitude of $w_t$ as a function of $t$, where $\{w_t\}$ is a sequence of output from algorithm 1. We shall see later on that a coarse bound for $w_t$ helps to proof Theorem 3.

**Lemma 8** (Exponential Growing Output). *Suppose $\mathcal{A}$ is an arbitrary OLO algorithm that guarantees regret $R_T^{\mathcal{A}}(0) = \sum_{t=1}^{T} \langle \mathbf{g}_t, \mathbf{w}_t \rangle \le \epsilon$ for all sequence $\{\mathbf{g}_t\}$ with $\|\mathbf{g}_t\| \le G$. Then it must hold that $\|\mathbf{w}_t\| \le \frac{\epsilon}{2G} 2^t$ for all $t$.*

*Proof.* We will first prove by contradiction that $\|\mathbf{w}_t\| \le \frac{\epsilon - \sum_{i=1}^{t-1}\langle \mathbf{g}_i, \mathbf{w}_i \rangle}{G}$ for all $t$ for all sequences $\{\mathbf{g}_t\}$. Suppose that there is some $t$ and sequence $\mathbf{g}_1, \ldots, \mathbf{g}_{t-1}$ such that $\|\mathbf{w}_t\| > \frac{\epsilon - \sum_{i=1}^{t-1}\langle \mathbf{g}_i, \mathbf{w}_i \rangle}{G}$. Then, consider $\mathbf{g}_t = G\frac{\mathbf{w}_t}{\|\mathbf{w}_t\|}$. Then we have:

$$R_t(0) = \sum_{i=1}^{t-1} \langle \mathbf{g}_i, \mathbf{w}_i \rangle + \langle \mathbf{g}_t, \mathbf{w}_t \rangle > \epsilon$$

which is a contradiction, and so $\|\mathbf{w}_t\| \le \frac{\epsilon - \sum_{i=1}^{t-1}\langle \mathbf{g}_i, \mathbf{w}_i \rangle}{G}$.

Now, if we define $H_t = \epsilon - \sum_{i=1}^{t} \langle \mathbf{g}_i, \mathbf{w}_i \rangle$, we have $\|\mathbf{w}_t\| \le \frac{H_{t-1}}{G}$. Therefore:

$$\begin{aligned} H_t &= H_{t-1} - \langle \mathbf{g}_t, \mathbf{w}_t \rangle \\ &\le 2H_{t-1} \\ &\le 2^t H_0 \\ &= \epsilon 2^t \end{aligned}$$

Thus, we have $\|\mathbf{w}_t\| \le \frac{H_{t-1}}{G} = \frac{\epsilon}{2G} 2^t$ as desired. $\square$

## B Cancellation for Gradients with Sub-exponential Noise

In this Section, we ultimately provide the proof for Theorem 3. We first show a few algebraic lemma followed by the property of the regularizer, Then we show the proof for Theorem 3 by combining different lemma with the outlines listed in Section 4.

**Lemma 9.** *For $x \ge 0, a > 0, p \ge 1$*

$$\frac{a}{(x+a)^{1-\frac{1}{p}}} \ge (x+a)^{\frac{1}{p}} - x^{\frac{1}{p}}$$

*Proof.*

$$x^{\frac{1}{p}}(x+a)^{1-\frac{1}{p}} \ge x^{\frac{1}{p}} x^{1-\frac{1}{p}} = x$$

rearrange

$$0 \ge x - x^{\frac{1}{p}}(x+a)^{1-\frac{1}{p}}$$

$$a \ge (x+a) - x^{\frac{1}{p}}(x+a)^{1-\frac{1}{p}}$$

divide both side by $(x+a)^{1-\frac{1}{p}}$, we complete the proof $\square$

**Lemma 10.** *For $x, a \geq 0, p \geq 1$:*

$$(x + a)^{1/p} \leq x^{1/p} + a^{1/p}$$

*Proof.*

$$x + a = (x^{1/p})^p + (a^{1/p})^p \leq (x^{1/p} + a^{1/p})^p$$

raise to the power of $1/p$ to complete the proof $\qquad \square$

**Lemma 11.** *For $\mathbf{x} \in \mathbb{R}^d$, if $\| \cdot \|_p$ is the $p-$norm:*

$$\frac{1}{d^{1/p}} \|\mathbf{x}\|_p \leq \|\mathbf{x}\|_\infty \leq \|\mathbf{x}\|_p$$

*Proof.* Clearly, it suffices to consider $\mathbf{x} = (x_1, \ldots, x_d)$ with $x_i \geq 0$ for all $i$. let $i_* = \mathrm{argmax}_i x_i$, then for all $a \geq 0$

$$\|\mathbf{x}\|_\infty = x_i^* = (x_i^{*p})^{1/p} \leq (x_i^{*p} + a)^{1/p}$$

setting $a = \sum_{i \neq i^*} x_i^p$, demonstrates the upper bound.

For the lower bound:

$$\frac{1}{d^{1/p}} \|\mathbf{x}\|_p = \frac{1}{d^{1/p}} \left( \sum_{i=1}^d x_i^p \right)^{1/p} \leq \frac{1}{d^{1/p}} (d x_i^{*p})^{1/p} = x_i^*$$

$\qquad \square$

**Lemma 12.** *For $x > 0$,*

$$\log(x + \exp(1)) \leq \max(0, \log(x)) + \exp(1)$$

*Proof.* For $x \in (0, 1]$, $\log(x) \leq 0$. Thus, the inequality holds since $\log(x + \exp(1)) \leq \log(1 + \exp(1)) \leq \exp(1)$.

For $x > 1$, we have $\log(x) > 0$. Let

$$h(x) = \log(x + \exp(1))$$
$$f(x) = \log(x) + \exp(1)$$

Taking derivatives,

$$h'(x) = 1/(x + \exp(1))$$
$$f'(x) = 1/x$$

Thus, $f'(x) > h'(x)$ for $x > 1$. Now, since $f(1) \geq h(1)$, we have $f(x) > h(x)$ for $x > 1$.

Combining both case we complete the proof. $\qquad \square$

**Lemma 13** (Cumulative Huber Loss). *Consider $r_t$ as in equation (5) (copied below for convience):*

$$r_t(w; c, p, \alpha_0) = \begin{cases} c \left( p|w| - (p-1)|w_t| \right) \frac{|w_t|^{p-1}}{\left( \sum_{i=1}^t |w_i|^p + \alpha_0^p \right)^{1-1/p}}, & |w| > |w_t| \\ c|w|^p \frac{1}{\left( \sum_{i=1}^t |w_i|^p + \alpha_0^p \right)^{1-1/p}}, & |w| \leq |w_t| \end{cases}$$

*Then, with fixed parameter $c, \alpha_0 > 0$, $p \geq 1$,*

$$\sum_{t=1}^T r_t(w_t) \geq c \left( \left( \sum_{t=1}^T |w_t|^p + \alpha_0^p \right)^{1/p} - \alpha_0 \right) \tag{10}$$

$$\sum_{t=1}^T r_t(u) \leq cp|u|T^{1/p} \left[ \left( \log \frac{T|u|^p + \alpha_0^p}{\alpha_0^p} \right)^{(p-1)/p} + 1 \right] \tag{11}$$

*Proof.* Define index set $A_T = \{t : |w_t| < |u|, t = 1, \cdots, T\}$, and let $n(A_T)$ be the cardinality of $A_T$. Let $S_t = \alpha_0^p + \sum_{i=1}^t |w_i|^p$. First, we show lower bound for $\sum_{t=1}^T r_t(w)$. Since $p \geq 1$,

$$\sum_{t=1}^T r_t(w_t) = c \sum_{t=1}^T \frac{|w_t|^p}{(\sum_{i=1}^t |w_i|^p + \alpha_0^p)^{1-1/p}}$$

$$= c \sum_{t=1}^T \frac{|w_t|^p}{(|w_t|^p + S_{t-1})^{1-1/p}}$$

use Lemma 9, set $a = |w_t|^p, x = S_{t-1}, a + x = S_t$

$$\geq c \sum_{t=1}^T (S_t^{1/p} - S_{t-1}^{1/p})$$

$$= c(S_T^{1/p} - \alpha_0)$$

$$= c\left( (\sum_{t=1}^T |w_t|^p + \alpha_0^p)^{1/p} - \alpha_0 \right)$$

Now, we upper bound of $\sum_t r_t(u)$. We partition the sum into two terms, and bound them individually:

$$\sum_{t=1}^T r_t(u) \leq cp|u| \underbrace{\sum_{\substack{t \leq T \\ |w_t| \leq |u|}} \frac{|w_t|^{p-1}}{(\sum_{i=1}^t |w_i|^p + \alpha_0^p)^{1-1/p}}}_{A} + c|u|^p \underbrace{\sum_{\substack{t \leq T \\ |w_t| > |u|}} \frac{1}{(\sum_{i=1}^t |w_i|^p + \alpha_0^p)^{1-1/p}}}_{B}$$

First, we bound $A$:

$$A \leq \sum_{\substack{t \leq T \\ |w_t| \leq |u|}} \frac{|w_t|^{p-1}}{\left( \sum_{\substack{i \leq t, \\ |w_i| \leq |u|}} |w_i|^p + \alpha_0^p \right)^{1-1/p}}$$

by Holder's inequality $\langle a, b \rangle \leq \|a\|_m \|b\|_n$, where $\frac{1}{m} + \frac{1}{n} = 1$. Set $m = p, n = \frac{p}{p-1}$.

$$\leq n(A_T)^{1/p} \left( \sum_{\substack{t \leq T \\ |w_t| \leq |u|}} \frac{|w_t|^p}{\sum_{\substack{i \leq t, \\ |w_i| \leq |u|}} |w_i|^p + \alpha_0^p} \right)^{(p-1)/p}$$

$$\leq n(A_T)^{1/p} \left( \int_{\alpha_0^p}^{\alpha_0^p + \sum_{t \in A_T} |w_t|^p} \frac{1}{x} dx \right)^{(p-1)/p}$$

$$= n(A_T)^{1/p} \left( \log \frac{\sum_{t \in A_T} |w_t|^p + \alpha_0^p}{\alpha_0^p} \right)^{(p-1)/p}$$

$$\leq n(A_T)^{1/p} \left( \log \frac{n(A_T)|u|^p + \alpha_0^p}{\alpha_0^p} \right)^{(p-1)/p}$$

$$\leq T^{1/p} \left( \log \frac{|u|^p T + \alpha_0^p}{\alpha_0^p} \right)^{(p-1)/p}$$

Now, we bound $B$:

$$B \leq \sum_{\substack{t \leq T \\ |w_t| > |u|}} \frac{1}{(\sum_{\substack{i \leq t, \\ |w_i| > |u|}} |w_i|^p + \alpha_0^p)^{1-1/p}}$$

$$\leq \sum_{\substack{t \leq T \\ |w_t| > |u|}} \frac{1}{[n(\{i \leq t : |w_i| > |u|\})|u|^p + \alpha_0^p]^{1-1/p}}$$

$$= \frac{1}{|u|^{p-1}} \sum_{\substack{t \leq T \\ |w_t| > |u|}} \frac{1}{\left[n(\{i \leq t : |w_i| > |u|\}) + (\frac{\alpha_0}{|u|})^p\right]^{1-1/p}}$$

$$\leq \frac{1}{|u|^{p-1}} \int_{(\alpha_0/|u|)^p}^{(\alpha_0/|u|)^p + (T - n(A_T))} \frac{1}{x^{1-1/p}} dx$$

$$= \frac{p}{|u|^{p-1}} \left[ (T - n(A_T)) + (\alpha_0/|u|)^p)^{1/p} - (\alpha_0/|u|) \right]$$

by Lemma 10, set $x = T - n(A_T), a = (\alpha_0/|u|)^p$

$$\leq \frac{p}{|u|^{p-1}} (T - n(A_T))^{1/p}$$

$$\leq \frac{p}{|u|^{p-1}} T^{1/p}$$

Combining $A$ and $B$:

$$\sum_{t=1}^{T} r_t(u) \leq cp|u|T^{1/p} \left[ \left( \log \frac{|u|^p T + \alpha_0^p}{\alpha_0^p} \right)^{(p-1)/p} + 1 \right]$$

$\square$

**Lemma 14.** *Consider the regularization function $\psi_t(w)$ defined in equation (6) with parameter $c_1, p_1, \alpha_1, c_2, p_2, \alpha_2$, displayed below for reference:*

$$\psi_t(w) = r_t(w; c_1, p_1, \alpha_1) + r_t(w; c_2, p_2, \alpha_2)$$

*When we set*

$$c_1 = 2\sigma \sqrt{\log\left(\frac{32}{\delta} \left[\log\left(2^{T+1}\right) + 2\right]^2\right)}, \quad c_2 = 32b \log\left(\frac{224}{\delta} \left[\log\left(1 + \frac{b}{\sigma} 2^{T+2}\right) + 2\right]^2\right),$$
$$p_1 = 2, \qquad p_2 = \log T, \qquad\qquad \alpha_1 = \epsilon/c_1, \qquad \alpha_2 = \epsilon\sigma/(4b(b+H))$$

*as defined Theorem 3, Then*

$$\sum_{t=1}^{T} \psi_t(w_t) \geq c_1 \sqrt{\sum_{t=1}^{T} |w_t|^2 + \alpha_1^2} + c_2 \max\left(\alpha_2, \max_{t \in \{1, \cdots, T\}} |w_t|\right) - \epsilon\left(1 + \frac{c_2\sigma}{4b(b+H)}\right)$$

$$\sum_{t=1}^{T} \psi_t(u) \leq 2c_1|u|\sqrt{T} \left[\sqrt{\log\left(\frac{T|u|^2 c_1^2}{\epsilon^2} + 1\right)} + 1\right] + 3c_2 p_2 |u| \left(\max\left(0, p_2\left(\log\frac{|u|}{\alpha_2} + 1\right)\right)\right) + 4)$$

*Proof.* The proof builds on Lemma 13. We show the algebra for $r_t$ with fixed tuple of parameters $(c_i, p_i, \alpha_i)$ for $i = 1, 2$, respectively. The main difference is due to the value of $p_i$.

For $i = 1$:

$$\sum_{t=1}^{T} r_t(w_t; c_1, p_1, \alpha_1) \geq c_1 \left(\sqrt{\sum_{t=1}^{T} |w_t|^2 + \alpha_1^2} - \alpha_1\right)$$

$$= c_1 \sqrt{\sum_{t=1}^{T} |w_t|^2 + \alpha_1^2} - \epsilon$$

By equation (11)

$$\sum_{t=1}^{T} r_t(u; c_1, p_1, \alpha_1) \leq 2c_1|u|\sqrt{T} \left[\sqrt{\log\frac{T|u|^2 + \alpha_1^2}{\alpha_1^2}} + 1\right]$$

$$= 2c_1|u|\sqrt{T}\left[\sqrt{\log\left(\frac{T|u|^2 c_1^2}{\epsilon^2}+1\right)}+1\right]$$

For $i=2$: by equation (10) and Lemma 11:

$$\sum_{t=1}^{T} r_t(w_t; c_2, p_2, \alpha_2) \geq c_2\left(\max\left(\alpha_2, \max_{t\in\{1,\cdots,T\}}|w_t|\right)-\alpha_2\right)$$

$$= c_2\max\left(\alpha_2, \max_{t\in\{1,\cdots,T\}}|w_t|\right)-\frac{\epsilon c_2\sigma}{4b(b+H)}$$

By equation (11)

$$\sum_{t=1}^{T} r_t(w_t; c_2, p_2, \alpha_2) \leq c_2 p_2 |u|\exp(1)\left[\left(\log\frac{T|u|^{p_2}+\alpha_2^{p_2}}{\alpha_2^{p_2}}\right)^{(p_2-1)/p_2}+1\right]$$

$$\leq c_2 p_2 |u|\exp(1)\left[\left(\log\frac{T|u|^{p_2}+\exp(1)\alpha_2^{p_2}}{\alpha_2^{p_2}}\right)^{(p_2-1)/p_2}+1\right]$$

$$\leq c_2 p_2 |u|\exp(1)\left[\left(\log\frac{T|u|^{p_2}+\exp(1)\alpha_2^{p_2}}{\alpha_2^{p_2}}\right)+1\right]$$

$$= c_2 p_2 |u|\exp(1)\left[\log\left(T\frac{|u|^{p_2}}{\alpha_2^{p_2}}+\exp(1)\right)+1\right]$$

Since $T\frac{|u|^{p_2}}{\alpha_2^{p_2}}>0$, invoke Lemma 12 by substituting $x=T\frac{|u|^{p_2}}{\alpha_2^{p_2}}$

$$\leq c_2 p_2 |u|\exp(1)\left(\max\left(0,\log\left(\frac{T|u|^{p_2}}{\alpha_{2p_2}}\right)\right)+\exp(1)+1\right)$$

$$= c_2 p_2 |u|\exp(1)\left(\max\left(0,p_2\log\left(\frac{\exp(1)|u|}{\alpha_2}\right)\right)+\exp(1)+1\right)$$

$$= c_2 p_2 |u|\exp(1)\left(\max\left(0,p_2\left(\log\frac{|u|}{\alpha_2}+1\right)\right)+\exp(1)+1\right)$$

$$\leq 3c_2 p_2 |u|\left(\max\left(0,p_2\left(\log\frac{|u|}{\alpha_2}+1\right)\right)+4\right)$$

Combining both cases of $i=1,2$, we complete the proof. $\qquad\square$

Now we are at the stage to prove Theorem 3. We restate the Theorem for reference, followed by the proof.

**Theorem 3.** *Suppose $\{g_t\}$ are stochastic subgradients such that $\mathbb{E}[g_t]\in\partial\ell_t(w_t)$, $|g_t|\leq b$ and $\mathbb{E}[g_t^2|w_t]\leq\sigma^2$ almost surely for all $t$. Set the following constants for $\psi_t(w)$ shown in equation (6) for any $0<\delta\leq 1$, $\epsilon>0$,*

$$c_1 = 2\sigma\sqrt{\log\left(\frac{32}{\delta}\left[\log\left(2^{T+1}\right)+2\right]^2\right)}, \quad c_2 = 32b\log\left(\frac{224}{\delta}\left[\log\left(1+\frac{b}{\sigma}2^{T+2}\right)+2\right]^2\right),$$

$$p_1 = 2, \qquad p_2 = \log T, \qquad\qquad \alpha_1 = \epsilon/c_1, \qquad \alpha_2 = \epsilon\sigma/(4b(b+H))$$

*where $H = c_1 p_1 + c_2 p_2$, $|\nabla\psi_t(w_t)|\leq H$. Then, with probability at least $1-\delta$, algorithm 1 guarantees*

$$R_T(u)\leq\tilde{O}\left[\epsilon\log\frac{1}{\delta}+|u|b\log\frac{1}{\delta}+|u|\sigma\sqrt{T\log\frac{1}{\delta}}\right]$$

*Proof.* The proof is a composition of concentration bounds and our Lemmas for the regularizers, following the outline in Section 4. Previously, we defined $\epsilon_t = \nabla\ell_t(\mathbf{w}_t)-g_t$. $|\epsilon_t|\leq 2b$ and $\mathbb{E}[\epsilon_t^2]\leq\sigma^2$.

**Step 1 :** We first derive a concentration bound for the NOISE term defined in equation (2). Notice that $\{|u|\epsilon_i\}$ is a martingale difference sequence. Then by Lemma 23, with probability at least $1 - \frac{\delta}{4}$,

$$\left| \sum_{t=1}^{T} u\epsilon_t \right| \leq 4|u|b \log \frac{8}{\delta} + |u|\sigma \sqrt{2T \log \frac{8}{\delta}} \tag{12}$$

Now, we coarsely bound the output $w_t$ from Algorithm 1. At each round $t$, $w_t$ is updated by solving

$$w_t = x_t - y_t \nabla \psi_t(w_t)$$
$$= \frac{x_t'}{b+H} - \frac{y_t'}{H(b+H)} \nabla \psi_t(w_t)$$

where $x_t', y_t'$ are outputs from some algorithm in which the regret at the origin is bounded by some positive $\epsilon$ with Lipschitz constant 1. By Lemma 8,

$$|x_t| \leq \frac{\epsilon}{2(b+H)} 2^t \qquad\qquad |y_t| \leq \frac{\epsilon}{2H(b+H)} 2^t$$

Now, define $k = b + H$. Then, by triangle inequality and $|\nabla \psi_t(w_t)| \leq H$

$$|w_t| \leq |x_t| + |y_t||\nabla \psi_t(w_t)| \leq \frac{\epsilon}{b+H} 2^t = \frac{\epsilon}{k} 2^t \tag{13}$$

Finally, $\{w_t \epsilon_t\}$ is a martingale difference sequence that satisfies:

$$\mathbb{E}[w_t^2 \epsilon_t^2 \mid g_1, \cdots, g_t] \leq |w_t|^2 \sigma^2$$
$$|w_t \epsilon_t| \leq 2|w_t|b$$

where $w_t$ depends on $g_1, \cdots, g_{t-1}$ only. Hence by Proposition 17 $\{w_t \epsilon_t\}$ is $(|w_t|\sigma, 4|w_t|b)$sub-exponential. Then we apply Theorem 18 by setting $\nu = \epsilon\sigma/k$ to obtain that with probability at least $1 - \frac{\delta}{4}$

$$\left| \sum_{t=1}^{T} w_t \epsilon_t \right| \leq 2 \sqrt{\sigma^2 \sum_{t=1}^{T} w_t^2 \log \left( \frac{32}{\delta} \left[ \log \left( \left[ \frac{k}{\epsilon} \sqrt{\sum_{t=1}^{T} w_t^2} \right]_1 \right) + 2 \right]^2 \right)}$$
$$+ 8 \max(\epsilon\sigma/k, 4b \max_{t \leq T} |w_t|) \log \left( \frac{224}{\delta} \left[ \log \left( \frac{\max(\epsilon\sigma/k, 4b\max_{t \leq T} |w_t|)}{\epsilon\sigma/k} \right) + 2 \right]^2 \right) \tag{14}$$

We now simplify the $\log\log$ term with a worst case upper bound of $|w_t|$. From equation (13), we have

$$\max i \leq t |w_i| \leq \frac{\epsilon}{k} 2^t, \qquad \sum_{i=1}^{t} w_i^2 \leq \frac{\epsilon^2}{k^2} \sum_{i=1}^{t} 4^i = \frac{\epsilon^2}{k^2} \frac{4(4^t - 1)}{3}$$

Hence

$$\log \left( \left[ \frac{k}{\epsilon} \sqrt{\sum_{i=1}^{t} w_i^2} \right]_1 \right) \leq \log \left( \left[ \sqrt{2 \cdot 4^T} \right]_1 \right) \leq \log \left( 2^{T+1} \right)$$

$$\log \left( \frac{\max(\epsilon\sigma/k, 4b \max_{t \leq T} |w_t|)}{\epsilon\sigma/k} \right) \leq \log \left( 1 + \frac{4bk \max_{t \leq T} |w_t|}{\epsilon\sigma} \right)$$
$$\leq \log \left( 1 + \frac{b}{\sigma} 2^{T+2} \right)$$

Notice that the double-logarithm in (14) is critical to ameliorate this exponential bound on $|w_t|$!

Substitute the above inequalities into equation (14), and combining with equation (12) by union bound, with probability at least $1 - \frac{\delta}{2}$:

$$\text{NOISE} \leq 4|u|b \log \frac{8}{\delta} + |u|\sigma \sqrt{2T \log \frac{8}{\delta}} + 2 \sqrt{\sigma^2 \sum_{t=1}^{T} w_t^2 \log \left( \frac{32}{\delta} \left[ \log \left( 2^{T+1} \right) + 2 \right]^2 \right)}$$

$$+ 8 \max(\epsilon\sigma/k, 4b \max_{t \leq T} |w_t|) \log\left(\frac{224}{\delta}\left[\log\left(1 + \frac{b}{\sigma}2^{T+2}\right) + 2\right]^2\right)$$

$$= 4|u|b\log\frac{8}{\delta} + |u|\sigma\sqrt{2T\log\frac{8}{\delta}} + 2\sqrt{\sigma^2\sum_{t=1}^{T}w_t^2\log\left(\frac{32}{\delta}\left[\log\left(2^{T+1}\right) + 2\right]^2\right)}$$

$$+ 32b\max(\epsilon\sigma/4kb, \max_{t \leq T}|w_t|)\log\left(\frac{224}{\delta}\left[\log\left(1 + \frac{b}{\sigma}2^{T+2}\right) + 2\right]^2\right)$$

$$= 4|u|b\log\frac{8}{\delta} + |u|\sigma\sqrt{2T\log\frac{8}{\delta}} + c_1\sqrt{\sum_{t=1}^{T}w_t^2} + c_2\max(\epsilon\sigma/4kb, \max_{t \leq T}|w_t|) \qquad (15)$$

**Step 2 :** Next, we derive a bound on $\sum_{t=1}^{T}\langle g_t, w_t - u\rangle$. Our approach builds upon the motivation sketched in equation (3). We define $R_T^A(u) = \sum_{t=1}^{T}\hat{\ell}_t(w_t) - \hat{\ell}_t(u)$. Notice that $R_T^A(u)$ can then be bounded by Theorem 2. Thus, we copy over equation (3) below, and apply Theorem 2 and Lemma 14 to bound the regret

$$\sum_{t=1}^{T}\langle g_t, w_t - u\rangle \leq R_T^A(u) - \sum_{t=1}^{T}\psi_t(w_t) + \sum_{t=1}^{T}\psi_t(u)$$

$$\leq 2\epsilon + |u|\left[\frac{3A}{2}\sqrt{N\max\left(0, \sum_{t=1}^{T}|g_t|^2 - |\nabla\psi_t(w_t)|^2\right)} + \left(A^2H + 2B(b + H)\right)N\right]$$

$$- c_1\sqrt{\sum_{t=1}^{T}|w_t|^2 + \alpha_1^2} - c_2\max\left(\alpha_2, \max_{t \in \{1,\cdots,T\}}|w_t|\right) + \epsilon\left(1 + \frac{c_2\sigma}{2b(b + H)}\right)$$

$$+ 2c_1|u|\sqrt{T}\left[\sqrt{\log\frac{T|u|^2 + \alpha_1^2}{\alpha_1^2}} + 1\right] + 3c_2p_2|u|\left(\max\left(0, p_2\left(\log\frac{|u|}{\alpha_2} + 1\right)\right) + 4\right) \qquad (16)$$

where $N = 1 + \log\left(\frac{(b+H)^2|u|^2T^C}{\epsilon^2} + 1\right)$ and $A, B, C$ are some positive constants.

**Step 3 :** As shown in equation (4), the regret is derived by combining equation (15) and (16). We observe that the martingale concentration from Step 1 will be cancelled by the negative regularization terms from Step 2 to complete the proof:

$$R_T(u) \leq \epsilon\left(3 + \frac{8\sigma}{b + H}\log\left(\frac{224}{\delta}\left[\log\left(1 + \frac{b}{\sigma}2^{T+2}\right) + 2\right]^2\right)\right)$$

$$+ |u|\left[4c_1(A^2 + B)N + \frac{3A}{2}\sqrt{N\max\left(0, \sum_{t=1}^{T}|g_t|^2 - |\nabla\psi_t(w_t)|^2\right)}\right]$$

$$+ |u|b\left[2BN + 4\log\frac{8}{\delta} + \frac{c_2\log T}{b}\left((A^2 + 2B)N + 3\left(\max\left(0, \log T\left(\log\frac{|u|}{\alpha_2} + 1\right)\right) + 4\right)\right)\right]$$

$$+ |u|\sqrt{T}\left[2c_1\left(\sqrt{\log\left(\frac{T|u|c_1^2}{\epsilon^2} + 1\right)} + 1\right) + \sigma\sqrt{2\log\frac{8}{\delta}}\right] \qquad (17)$$

The above holds for probability at least $1 - \frac{\delta}{2}$.

**Step 4:** For the final statement, we must remove the random quantity $\sum_{t=1}^{T}g_t^2$ appearing in the bound. Fortunately, this is achievable via a relatively straightforward application of Bernstein-style

bounds. In particular, by Lemma 24, with probability at least $1 - \delta/2$

$$\sum_{t=1}^{T} |g_t|^2 \leq \frac{3}{2}T\sigma^2 + \frac{5}{3}b^2 \log \frac{2}{\delta}$$

Thus, further upper bound equation 17 by union bound, we have with probability at least $1 - \delta$,

$$
\begin{aligned}
R_T(u) \leq{} & \epsilon \left( 3 + \frac{8\sigma}{b+H} \log \left( \frac{224}{\delta} \left[ \log \left( 1 + \frac{b}{\sigma}2^{T+2} \right) + 2 \right]^2 \right) \right) \\
& + |u| \left[ 4c_1(A^2+B)N + \frac{3A}{2}\sqrt{N\left(\frac{3}{2}T\sigma^2 + \frac{5}{3}b^2 \log \frac{2}{\delta}\right)} \right] \\
& + |u|b \left[ 2BN + 4\log\frac{8}{\delta} + \frac{c_2 \log T}{b}\left( (A^2+2B)N + 3\left(\max\left(0, \log T\left(\log\frac{|u|}{\alpha_2}+1\right)\right)+4\right)\right) \right] \\
& + |u|\sqrt{T} \left[ 2c_1 \left( \sqrt{\log\left(\frac{T|u|c_1^2}{\epsilon^2}+1\right)}+1 \right) + \sigma\sqrt{2\log\frac{8}{\delta}} \right]
\end{aligned}
\tag{18}
$$

$\square$

## C    Gradient Clipping for Heavy-tailed Gradients

First, we show the property of truncated heavy-tailed gradients followed by the proof of Theorem 4. These elementary facts can be found in Zhang et al. [2020], but we reproduce the proofs for completeness.

**Lemma 15** (Clipped Gradient Properties). *Suppose $\mathbf{g}_t$ is heavy-tailed random vector, $\|\mathbb{E}[\mathbf{g}_t]\| \leq G$, $\mathbb{E}[\|\mathbf{g}_t - \mathbb{E}[\mathbf{g}_t]\|^{\mathfrak{p}}] \leq \sigma^{\mathfrak{p}}$ for some $\mathfrak{p} \in (1,2]$ and $\sigma \leq \infty$. Define truncated gradient $\hat{\mathbf{g}}_t$ with a positive clipping parameter $\tau$:*

$$\hat{\mathbf{g}}_t = \frac{\mathbf{g}_t}{\|\mathbf{g}_t\|}\min(\tau, \|\mathbf{g}_t\|)$$

*Let $\boldsymbol{\mu} = \mathbb{E}[\mathbf{g}_t]$. Then:*

$$\|\mathbb{E}[\hat{\mathbf{g}}_t] - \boldsymbol{\mu}\| \leq \frac{2^{\mathfrak{p}-1}(\sigma^{\mathfrak{p}}+G^{\mathfrak{p}})}{\tau^{\mathfrak{p}-1}}$$
$$\mathbb{E}[\|\hat{\mathbf{g}}_t\|^2] \leq 2^{\mathfrak{p}-1}\tau^{2-\mathfrak{p}}(\sigma^{\mathfrak{p}}+G^{\mathfrak{p}})$$

*Proof.* By Jensen's inequality

$$
\begin{aligned}
\|\mathbb{E}[\hat{\mathbf{g}}_t] - \boldsymbol{\mu}\| & \leq \mathbb{E}[\|\hat{\mathbf{g}}_t - \mathbf{g}_t\|] \\
& \leq \mathbb{E}[\|\mathbf{g}_t\|\mathbf{1}[\|\mathbf{g}_t\| \geq \tau]] \\
& \leq \mathbb{E}[\|\mathbf{g}_t\|^{\mathfrak{p}}/\tau^{\mathfrak{p}-1}] \\
& \leq \mathbb{E}[(\|\mathbf{g}_t - \boldsymbol{\mu}\| + \|\boldsymbol{\mu}\|)^{\mathfrak{p}}/\tau^{\mathfrak{p}-1}] \\
& = \frac{2^{\mathfrak{p}}}{\tau^{\mathfrak{p}-1}}\mathbb{E}[(\frac{1}{2}\|\mathbf{g}_t - \boldsymbol{\mu}\| + \frac{1}{2}\|\boldsymbol{\mu}\|)^2] \\
& \leq \frac{2^{\mathfrak{p}-1}}{\tau^{\mathfrak{p}-1}}\left(\mathbb{E}[\|\mathbf{g}_t - \boldsymbol{\mu}\|^2] + \mathbb{E}[\|\boldsymbol{\mu}\|^2]\right) \\
& \leq \frac{2^{\mathfrak{p}-1}(\sigma^{\mathfrak{p}}+G^{\mathfrak{p}})}{\tau^{\mathfrak{p}-1}}
\end{aligned}
$$

The second last inequality was due to convexity and linearity of expectation. In term of the variance, the algebra is similar:

$$
\begin{aligned}
\mathbb{E}[\|\hat{\mathbf{g}}_t\|^2] & \leq \mathbb{E}[\|\mathbf{g}_t\|^{\mathfrak{p}}\tau^{2-\mathfrak{p}}] \\
& \leq \mathbb{E}[(\|\mathbf{g}_t - \boldsymbol{\mu}\| + \|\boldsymbol{\mu}\|)^{\mathfrak{p}}\tau^{2-\mathfrak{p}}]
\end{aligned}
$$

$$= 2^{\mathfrak{p}}\tau^{2-\mathfrak{p}}\,\mathbb{E}[(\tfrac{1}{2}\|\mathbf{g}_t - \boldsymbol{\mu}\| + \tfrac{1}{2}\|\boldsymbol{\mu}\|)^{\mathfrak{p}}]$$

$$\leq 2\,(\mathbb{E}[\|\mathbf{g}_t - \boldsymbol{\mu}\|^{\mathfrak{p}}] + \mathbb{E}[\|\boldsymbol{\mu}\|^{\mathfrak{p}}])$$

$$\leq 2^{\mathfrak{p}-1}\tau^{2-\mathfrak{p}}(\sigma^{\mathfrak{p}} + G^{\mathfrak{p}})$$

$\square$

We now restate Theorem 4 followed by its proof.

**Theorem 4.** *Suppose $\{g_t\}$ are heavy-tailed stochastic gradient such that $\mathbb{E}[g_t] \in \partial \ell_t(w_t)$, $|\mathbb{E}[g_t]| \leq G$, $\mathbb{E}[|g_t - \mathbb{E}[g_t]|^{\mathfrak{p}}] \leq \sigma^{\mathfrak{p}}$ for some $\mathfrak{p} \in (1, 2]$. If we set $\tau = T^{1/\mathfrak{p}}(\sigma^{\mathfrak{p}} + G^{\mathfrak{p}})^{1/\mathfrak{p}}$ then with probability at least $1 - \delta$, Algorithm 2 guarantees:*

$$R_T(u) \leq \tilde{O}\left[\epsilon \log \frac{1}{\delta} + |u|T^{1/\mathfrak{p}}(\sigma + G)\log \frac{T}{\delta}\log \frac{|u|T}{\epsilon}\right]$$

*Proof.* We copy over $\phi(w)$ and regret formula in equation (8) and (9) from section 5 here, as the analysis will be following the cancellation-by-regularization strategy described in section 5.

$$\phi(w) = 2^{\mathfrak{p}-1}(\sigma^{\mathfrak{p}} + G^{\mathfrak{p}})|w|/\tau^{\mathfrak{p}-1} \qquad\qquad \hat{\ell}_t(w) = \langle \mathbb{E}[\hat{g}_t], w - u\rangle + \phi(w)$$

$$R_T(u) \leq \underbrace{\sum_{t=1}^{T}\left(\langle \nabla \ell_t(w_t) - \mathbb{E}[\hat{g}_t], w_t - u\rangle - \phi(w_t) + \phi(u)\right)}_{D} + \underbrace{\sum_{t=1}^{T}\hat{\ell}_t(w_t) - \hat{\ell}_t(u)}_{E}$$

We split the regret into two parts. The term $D$ is controlled by cancellation-by-regularization through careful choice of $\phi_t(w)$ and clipping parameter $\tau$. Term $E$ is controlled in high probability through Algorithm 1 (which uses a *different* cancellation-by-regularization strategy) by sending $\hat{g}_t + \nabla\phi(w_t)$ as the $t^{th}$ subgradient. Specifically, we can view $\hat{g}_t + \nabla\phi(w_t)$ as a sub-exponential and bounded noisy gradient and $\mathbb{E}[\hat{g}_t + \nabla\phi(w_t)] = \nabla\hat{\ell}_t(w_t)$, so that Theorem 3 provides a high probability bound for term $E$.

**First, we bound $D$,** and show it's independent of $|w_t|$

$$D \leq \sum_{t=1}^{T}|\nabla\ell_t(w_t) - \mathbb{E}[\hat{g}_t]|(|w_t| + |u|) - 2^{\mathfrak{p}-1}(\sigma^{\mathfrak{p}} + G^{\mathfrak{p}})/\tau^{\mathfrak{p}-1}\sum_{t=1}^{T}|w_t| + 2^{\mathfrak{p}-1}(\sigma^{\mathfrak{p}} + G^{\mathfrak{p}})T|u|/\tau^{\mathfrak{p}-1}$$

since $\nabla\ell_t(w_t) = \mathbb{E}[g_t]$, by Lemma 15, $|\nabla\ell_t(w_t) - \mathbb{E}[\hat{g}_t]| \leq 2^{\mathfrak{p}-1}(\sigma^{\mathfrak{p}} + G^{\mathfrak{p}})/\tau^{\mathfrak{p}-1}$.

$$\leq 2^{\mathfrak{p}}T|u|(\sigma^{\mathfrak{p}} + G^{\mathfrak{p}})/\tau^{\mathfrak{p}-1}$$

set $\tau = T^{1/\mathfrak{p}}(\sigma^{\mathfrak{p}} + G^{\mathfrak{p}})^{1/\mathfrak{p}}$

$$= 2^{\mathfrak{p}}|u|T^{1/\mathfrak{p}}(\sigma^{\mathfrak{p}} + G^{\mathfrak{p}})^{1/\mathfrak{p}}$$

$$\leq 4|u|T^{1/\mathfrak{p}}(\sigma^{\mathfrak{p}} + G^{\mathfrak{p}})^{1/\mathfrak{p}}$$

**Now we bound $E$ in high probability** with $\tau = T^{1/\mathfrak{p}}(\sigma^{\mathfrak{p}} + G^{\mathfrak{p}})^{1/\mathfrak{p}}$. We sometimes will substitute the value of $\tau$ and sometimes leave it as it is during the derivation for convenience. Define the noise as $\epsilon_t$,

$$\epsilon_t = \nabla\hat{\ell}_t(w_t) - (\hat{g}_t + \nabla\phi(w_t)) = \mathbb{E}[\hat{g}_t] - \hat{g}_t$$

From the definition of gradient clipping, $|\hat{g}_t| \leq \tau$. Also by Lemma 15,

$$\mathbb{E}[\hat{g}_t^2|w_t] \leq 2\tau^{2-\mathfrak{p}}(\sigma^{\mathfrak{p}} + G^{\mathfrak{p}}) = 2\tau^2 T^{-1}$$

Hence term $E$ can be bounded by Theorem 3, where we set the following constants for Algorithm 1,

$$c_1 = 2\tau\sqrt{\frac{2}{T}\log\left(\frac{32}{\delta}[\log(2^{T+1}) + 2]^2\right)}, \quad c_2 = 32\tau\log\left(\frac{224}{\delta}\left[\log\left(1 + \sqrt{T}2^{T+5/2}\right) + 2\right]^2\right),$$

$$p_1 = 2, \qquad p_2 = \log T, \qquad\qquad \alpha_1 = \epsilon/c_1, \qquad \alpha_2 = (\sqrt{2}\epsilon)/(4\sqrt{T}(\tau + H))$$

where $H = c_1 p_1 + c_2 p_2$. Let $N = 1 + \log\left(\frac{(\tau+H)^2|u|^2 T^C}{\epsilon^2} + 1\right)$. Then by equation (17), with probability at least $1 - \frac{\delta}{2}$ for some positive $A, B, C$,

$$E \leq \epsilon\left(3 + \frac{8\tau\sqrt{2/T}}{\tau + H}\log\left(\frac{224}{\delta}\left[\log\left(1 + \sqrt{T}2^{T+5/2}\right) + 2\right]^2\right)\right)$$

$$+ |u|\left[4c_1(A^2 + B)N + \frac{3A}{2}\sqrt{N\max\left(0, \sum_{t=1}^{T}|g_t|^2 - |\nabla\psi_t(w_t)|^2\right)}\right]$$

$$+ |u|\tau\left[2BN + 4\log\frac{8}{\delta} + \frac{c_2\log T}{\tau}\left((A^2 + 2B)N + 3\left(\max\left(0, \log T\left(\log\frac{|u|}{\alpha_2} + 1\right)\right) + 4\right)\right)\right]$$

$$+ |u|\sqrt{T}\left[2c_1\left(\sqrt{\log\left(\frac{T|u|c_1^2}{\epsilon^2} + 1\right)} + 1\right) + 2\tau\sqrt{\frac{1}{T}\log\frac{8}{\delta}}\right]$$

Combining $D$, $E$ and substitute $\tau$ when convenient and group in terms of the product of $|u|$ with $T$ to some power of $\mathfrak{p}$

$$R_T(\mathbf{u}) \leq \epsilon\left(3 + \frac{8\tau\sqrt{2/T}}{\tau + H}\log\left(\frac{224}{\delta}\left[\log\left(1 + \sqrt{T}2^{T+5/2}\right) + 2\right]^2\right)\right)$$

$$+ |u|\frac{3A}{2}\sqrt{N\max\left(0, \sum_{t=1}^{T}|\hat{g}_t + \nabla\phi(w_t)|^2 - |\nabla\psi_t(w_t)|^2\right)}$$

$$+ |u|T^{1/\mathfrak{p}}(\sigma^\mathfrak{p} + G^\mathfrak{p})^{1/\mathfrak{p}}\left[2BN + 2\sqrt{\log\frac{8}{\delta}} + 4\log\frac{8}{\delta}\right.$$

$$+ 4\sqrt{2\log\left(\frac{32}{\tau}[\log(2^{T+1}) + 2]^2\right)}\left(\frac{1}{\sqrt{T}}(A^2 + 2B)N + \sqrt{\log\left(\frac{T|u|c_1^2}{\epsilon^2} + 1\right)} + 1\right)$$

$$+ \left.\frac{c_2\log T}{\tau}\left(2(A^2 + B)N + 3\max\left(\max\left(0, \log T\left(\log\frac{|u|}{\alpha_2} + 1\right)\right) + 4\right) + 4\right)\right]$$

$$\tag{19}$$

For the final statement, notice that although $\hat{g}_t$ is a random quantity, we can bound $\sum_{t=1}^{T}|\hat{g}_t|^2$ with high probability. By Lemma 15, $\mathbb{E}[\hat{g}_t^2] \leq 2^{\mathfrak{p}-1}\tau^{2-\mathfrak{p}}(\sigma^\mathfrak{p} + G^\mathfrak{p})$ and note $|\hat{g}_t| \leq \tau$. Thus by Lemma 24, with probability at least $1 - \delta/2$

$$\sum_{t=1}^{T}|\hat{g}_t|^2 \leq 3T2^{\mathfrak{p}-2}\tau^{2-\mathfrak{p}}(\sigma^\mathfrak{p} + G^\mathfrak{p}) + 2\tau^2\log\frac{2}{\delta}$$

$$\leq \tau^2(3 + 2\log\frac{2}{\delta})$$

Finally, since equation (19) holds for probability as least $1 - \frac{\delta}{2}$, we further upperbound $|\hat{g}_t|^2$ by union bound for our final regret guarantee with probability at least $1 - \delta$

$$R_T(u) \leq \epsilon\left(3 + \frac{8\tau\sqrt{2/T}}{\tau + H}\log\left(\frac{224}{\delta}\left[\log\left(1 + \sqrt{T}2^{T+5/2}\right) + 2\right]^2\right)\right)$$

$$+ |u|\frac{3A}{2}\sqrt{N\max\left(0, \tau^2(3 + 2\log\frac{2}{\delta}) + \sum_{t=1}^{T}|\nabla\phi(w_t)|^2 - |\nabla\psi_t(w_t)|^2\right)}$$

$$+ |u|T^{1/\mathfrak{p}}(\sigma^\mathfrak{p} + G^\mathfrak{p})^{1/\mathfrak{p}}\left[2BN + 2\sqrt{\log\frac{8}{\delta}} + 4\log\frac{8}{\delta}\right.$$

$$+ 4\sqrt{2\log\left(\frac{32}{\delta}\left[\log\left(2^{T+1}\right)+2\right]^2\right)}\left(\frac{1}{\sqrt{T}}(A^2+2B)N+\sqrt{\log\left(\frac{T|u|c_1^2}{\epsilon^2}+1\right)}+1\right)$$

$$+\frac{c_2\log T}{\tau}\left(2(A^2+B)N+3\max\left(0,\log T\log\left(\frac{3|u|}{\alpha_2}\right)\right)+4\right)\Bigg]$$

where $|\nabla\phi(w_t)|\leq 2^{\mathfrak{p}-1}T^{\frac{1}{\mathfrak{p}}-1}(\sigma^{\mathfrak{p}}+G^{\mathfrak{p}})^{\frac{1}{\mathfrak{p}}}$ $\hfill\square$

## D  Dimension-free Gradient Clipping for Heavy-tailed Gradients

**Lemma 16.** *(Unit Ball Domain Algorithm High Probability) Suppose $\{\mathbf{g}_t\}$ is a sequence of heavy-tailed stochastic gradient vectors such that $\mathbb{E}[\|\mathbf{g}_t\|]\leq G$, $\mathbb{E}[\|\mathbf{g}_t-\mathbb{E}[\mathbf{g}_t]\|^{\mathfrak{p}}]\leq\sigma^{\mathfrak{p}}$ for some $\mathfrak{p}\in(1,2]$. Let $\hat{\mathbf{g}}_t$ be the clipped gradient $\hat{\mathbf{g}}_t=\mathbf{g}_t/\|\mathbf{g}_t\|\min(\tau,\|\mathbf{g}_t\|)$, where $\tau$ is set as $T^{1/\mathfrak{p}}(\sigma^{\mathfrak{p}}+G^{\mathfrak{p}})^{1/\mathfrak{p}}$. The constrained domain on unit ball ensures $\|\mathbf{w}_t\|,\|\mathbf{u}\|\leq 1$. Then with probability at least $1-\delta$, algorithm 3 guarantees*

$$R_T^{nd}(\mathbf{u})\leq\sum_{t=1}^{T}\langle\mathbb{E}[\mathbf{g}_t],\mathbf{w}_t-\mathbf{u}\rangle$$

$$\leq T^{1/\mathfrak{p}}(\sigma^{\mathfrak{p}}+G^{\mathfrak{p}})^{1/\mathfrak{p}}\left(\frac{1}{2}\|\mathbf{u}\|^2+\left(\frac{3}{2}+\frac{20}{3}\log\frac{2}{\delta}\right)+15\sqrt{\log\frac{160}{\delta}}\right.$$

$$\left.+184\log\left(\frac{448}{\delta}\left[\log\left(2+\frac{16}{\tau}\right)+1\right]^2\right)+4\right)$$

*Proof.* The analysis follows similar to the $1d$ analysis as seen in Theorem 4. We run a standard Follow-the-regularized-leader (FTRL) algorithm with $L_2$ regularization on clipped gradient $\hat{\mathbf{g}}_t$ instead of the true gradient $\nabla\ell_t(\mathbf{w}_t)$. A 'bias' term was introduced by $\mathbb{E}[\mathbf{g}_t]-\mathbb{E}[\hat{\mathbf{g}}_t]$ which can be regulated to be sublinear by the clipping parameter $\tau$. In addition, a 'noise' term due to $\mathbb{E}[\hat{\mathbf{g}}_t]-\hat{\mathbf{g}}_t$ can be bounded with high probability. We decompose the regret and label the corresponding parts below,

$$R_T^{nd}(\mathbf{u})=\sum_{t=1}^{T}\langle\nabla\ell_t(\mathbf{w}_t),\mathbf{w}_t-\mathbf{u}\rangle$$

$$=\sum_{t=1}^{T}\langle\hat{\mathbf{g}}_t,\mathbf{w}_t-\mathbf{u}\rangle+\sum_{t=1}^{T}\langle\nabla\ell_t(\mathbf{w}_t)-\mathbb{E}[\hat{\mathbf{g}}_t],\mathbf{w}_t-\mathbf{u}\rangle+\sum_{t=1}^{T}\langle\mathbb{E}[\hat{\mathbf{g}}_t]-\hat{\mathbf{g}}_t,\mathbf{w}_t-\mathbf{u}\rangle$$

$$\leq\underbrace{\|\sum_{t=1}^{T}\langle\hat{\mathbf{g}}_t,\mathbf{w}_t-\mathbf{u}\rangle\|}_{\text{FTRL}}+\underbrace{\sum_{t=1}^{T}\langle\nabla\ell_t(\mathbf{w}_t)-\mathbb{E}[\hat{\mathbf{g}}_t],\mathbf{w}_t-\mathbf{u}\rangle}_{\text{'bias'}}+\underbrace{\|\sum_{t=1}^{T}\langle\mathbb{E}[\hat{\mathbf{g}}_t]-\hat{\mathbf{g}}_t,\mathbf{w}_t-\mathbf{u}\rangle\|}_{\text{'noise'}}$$

Now we will bound the regret in three step.

**Step 1 :** For the part controlled by a FTRL algorithm with fixed $L_2$ regularization weight $\eta$ (see Corollary 7.8 in Orabona [2019])

$$\sum_{t=1}^{T}\langle\hat{\mathbf{g}}_t,\mathbf{w}_t-\mathbf{u}\rangle\leq\frac{1}{2\eta}\|\mathbf{u}\|^2+\frac{\eta}{2}\sum_{t=1}^{T}\|\hat{\mathbf{g}}_t\|^2$$

Since $\|\hat{\mathbf{g}}_t\|\leq\tau$, and by Lemma 15, $\mathbb{E}[\|\hat{\mathbf{g}}_t\|^2]\leq 2^{\mathfrak{p}-1}\tau^{2-\mathfrak{p}}(\sigma^{\mathfrak{p}}+G^{\mathfrak{p}})\leq 2\tau/T$. By Proposition 17, $\mathbf{g}_t$ is $(\tau\sqrt{2/T},2\tau)$ sub-exponential. Hence by Lemma 24, with probability at least $1-\delta$

$$\leq\frac{1}{2\eta}\|\mathbf{u}\|^2+\frac{\eta}{2}\left(3\tau^2+\frac{20}{3}\tau^2\log\frac{1}{\delta}\right)$$

set $\eta=1/\tau$

$$\leq\frac{\tau}{2}\|\mathbf{u}\|^2+\tau\left(\frac{3}{2}+\frac{20}{3}\log\frac{1}{\delta}\right)$$

**Step 2 :** For the 'bias' term, note $\nabla \ell_t(\mathbf{w}_t) = \mathbb{E}[\mathbf{g}_t]$, by Lemma 15, $\| \mathbb{E}[\hat{\mathbf{g}}_t] - \mathbb{E}[\mathbf{g}_t]\| \leq \frac{2^{\mathfrak{p}-1}(\sigma^{\mathfrak{p}}+G^{\mathfrak{p}})}{\tau^{\mathfrak{p}-1}}$, and the constrained domain suggests $\|\mathbf{w}_t - \mathbf{u}\| \leq 2$

$$\sum_{t=1}^{T} \langle \nabla \ell_t(\mathbf{w}_t) - \mathbb{E}[\hat{\mathbf{g}}_t], \mathbf{w}_t - \mathbf{u} \rangle \leq \frac{2^{\mathfrak{p}} T}{\tau^{\mathfrak{p}-1}} (\sigma^{\mathfrak{p}} + G^{\mathfrak{p}})$$

**Step 3 :** For a high probability bound for the 'noise' term, let $X_t = \langle \mathbb{E}[\hat{\mathbf{g}}_t] - \hat{\mathbf{g}}_t, \mathbf{w}_t - \mathbf{u} \rangle$, hence $\{X_t\}$ is a vector valued MDS adapted to filtration $\mathcal{F}_t$ with the following bound almost surely,

$$\begin{aligned}
X_t &\leq (\mathbb{E}[\hat{\mathbf{g}}_t] + \|\hat{\mathbf{g}}_t\|\|)\mathbf{w}_t - \mathbf{u}\| \\
&\leq 2\tau \|\mathbf{w}_t - \mathbf{u}\|
\end{aligned}$$

$$\mathbb{E}[\|X_t\|^2 \mid \mathcal{F}_t] \leq \|\mathbf{w}_t - \mathbf{u}\|^2 \, \mathbb{E}[\|\hat{\mathbf{g}}_t\|^2 \mid \mathcal{F}_t]$$

by Lemma 15

$$\begin{aligned}
&\leq \|\mathbf{w}_t - \mathbf{u}\|^2 2^{\mathfrak{p}-1} \tau^{2-\mathfrak{p}} (\sigma^{\mathfrak{p}} + G^{\mathfrak{p}}) \\
&\leq 2\|\mathbf{w}_t - \mathbf{u}\|^2 T^{2/\mathfrak{p}-1} (\sigma^{\mathfrak{p}} + G^{\mathfrak{p}})^{2/\mathfrak{p}} \\
&= 2\|\mathbf{w}_t - \mathbf{u}\|^2 \tau^2 T^{-1}
\end{aligned}$$

and both bounds are $\mathcal{F}_{t-1}$ measurable. By proposition 17 $\{\langle \mathbb{E}[\hat{\mathbf{g}}_t] - \hat{\mathbf{g}}_t, \mathbf{w}_t - \mathbf{u} \rangle\}$ is $\{\sqrt{2}\|\mathbf{w}_t - \mathbf{u}\|\tau T^{-1/2}, 4\tau\|\mathbf{w}_t - \mathbf{u}\|\}$ sub-exponential noise. Use Theorem 19 and set $\nu = \tau$, with probability at least $1 - \delta$,

$$\begin{aligned}
\left\| \sum_{t=1}^{T} X_t \right\| &\leq 5\sqrt{2\frac{\tau^2}{T}\sum_{t=1}^{T}\|\mathbf{w}_t - \mathbf{u}\|^2 \log\left(\frac{16}{\delta}\left[\log\left(\left[\sqrt{\frac{2}{T}\sum_{t=1}^{T}\|\mathbf{w}_t - \mathbf{u}\|^2}\right]_1\right) + 1\right]^2\right)} \\
&\quad + 23\max(\tau, 4\tau \max_{t \leq T}\|\mathbf{w}_t - \mathbf{u}\|)\log\left(\frac{224}{\delta}\left[\log\left(2\max(1, \frac{4}{\tau}\max_{t\leq T}\|\mathbf{w}_t - \mathbf{u}\|)\right) + 1\right]^2\right) \\
&\leq 15\tau\sqrt{\log\frac{80}{\delta}} + 184\tau\log\left(\frac{224}{\delta}\left[\log\left(2 + \frac{16}{\tau}\right) + 1\right]^2\right) \\
&= T^{1/\mathfrak{p}}(\sigma^{\mathfrak{p}} + G^{\mathfrak{p}})^{1/\mathfrak{p}}\left(15\sqrt{\log\frac{80}{\delta}} + 184\log\left(\frac{224}{\delta}\left[\log\left(2 + \frac{16}{\tau}\right) + 1\right]^2\right)\right)
\end{aligned}$$

**Composition :** Combining the high probability bound from step 1 and 3 by union bound and a deterministic bound from step 2, with probability at least $1 - \delta$, we have the following regret guarantee,

$$\begin{aligned}
R_T^{nd}(\mathbf{u}) &\leq \frac{\tau}{2}\|\mathbf{u}\|^2 + \tau\left(\frac{3}{2} + \frac{20}{3}\log\frac{2}{\delta}\right) + \frac{2^{\mathfrak{p}}T}{\tau^{\mathfrak{p}-1}}(\sigma^{\mathfrak{p}} + G^{\mathfrak{p}}) \\
&\quad + T^{1/\mathfrak{p}}(\sigma^{\mathfrak{p}} + G^{\mathfrak{p}})^{1/\mathfrak{p}}\left(15\sqrt{\log\frac{160}{\delta}} + 184\log\left(\frac{448}{\delta}\left[\log\left(2 + \frac{16}{\tau}\right) + 1\right]^2\right)\right)
\end{aligned}$$

substitute $\tau = T^{1/\mathfrak{p}}(\sigma^{\mathfrak{p}} + G^{\mathfrak{p}})^{1/\mathfrak{p}}$ and group terms by factorizing some power of $T$

$$\begin{aligned}
&\leq T^{1/\mathfrak{p}}(\sigma^{\mathfrak{p}} + G^{\mathfrak{p}})^{1/\mathfrak{p}}\left(\frac{1}{2}\|\mathbf{u}\|^2 + \left(\frac{3}{2} + \frac{20}{3}\log\frac{2}{\delta}\right) + 15\sqrt{\log\frac{160}{\delta}}\right. \\
&\quad \left. + 184\log\left(\frac{448}{\delta}\left[\log\left(2 + \frac{16}{\tau}\right) + 1\right]^2\right) + 4\right)
\end{aligned}$$

$\square$

We restate Theorem 5 for reference followed by its proof

**Theorem 5.** *Suppose that for all $t$, $\{\mathbf{g}_t\}$ are heavy-tailed stochastic subgradients satisfying $\mathbb{E}[\mathbf{g}_t] \in \partial\ell_t(\mathbf{w}_t)$, $\|\mathbb{E}[\mathbf{g}_t]\| \le G$ and $\mathbb{E}[\|\mathbf{g}_t - \mathbb{E}[\mathbf{g}_t]\|^{\mathfrak{p}}] \le \sigma^{\mathfrak{p}}$ for some $\mathfrak{p} \in (1,2]$. Then, with probability at least $1 - \delta$, Algorithm 4 guarantees*

$$R_T(\mathbf{u}) = \sum_{t=1}^{T} \ell_t(\mathbf{w}_t) - \ell_t(\mathbf{u}) \le \tilde{O}\left[\epsilon \log \frac{1}{\delta} + \|\mathbf{u}\| T^{1/\mathfrak{p}}(\sigma + G) \log \frac{T}{\delta} \log \frac{\|\mathbf{u}\| T}{\epsilon}\right]$$

*Proof.* This result relies on the reduction from dimension-free learning to 1d learning presented in Theorem 2 of Cutkosky and Orabona [2018]. This result implies that the regret 4 can be be bounded as:

$$\sum_{t=1}^{T} \ell_t(\mathbf{w}_t) - \ell_t(\mathbf{u}) \le R_T^{1d}(\|\mathbf{u}\|) + \|\mathbf{u}\| R_T^{nd}(\mathbf{u}/\|\mathbf{u}\|)$$

where

$$R_T^{1d}(\|u\|) = \sum_{t=1}^{T} \langle g_t^{1d}, x_t - \|u\|\rangle$$

$$R_T^{nd}(\mathbf{u}/\|\mathbf{u}\|) = \sum_{t=1}^{T} \langle \mathbf{g}_t, \mathbf{v}_t - \mathbf{u}/\|\mathbf{u}\|\rangle$$

First, by setting Algorithm 2 as $\mathcal{A}^{1d}$ is , Theorem 4 provides a bound for $R_T^{1d}(\|\mathbf{u}\|)$ with appropriate parameters set in the theorem. We run $\mathcal{A}^{1d}$ on subgradients $g_t = \langle \mathbf{g}_t, \mathbf{v}_t\rangle$, and $\|\mathbf{v}_t\| \le 1$. We show $|\mathbb{E}[g_t]|$ and $\mathbb{E}[\|g_t - \mathbb{E}[g_t]\|^{\mathfrak{p}}]$ are bounded. Notice that $g_t$ only depends on $\mathbf{g}_1, \cdots, \mathbf{g}_{t-1}$. Thus, by tower rule, $\mathbb{E}[g_t] = \mathbb{E}[\langle \mathbf{g}_t, \mathbf{v}_t\rangle] = \mathbb{E}[\langle \mathbb{E}[\mathbf{g}_t], \mathbf{v}_t\rangle \mid \mathbf{g}_1, \cdots, \mathbf{g}_{t-1}]$. Then, since $\|\mathbf{v}_t\| \le 1$ we have $|\langle \mathbb{E}[\mathbf{g}_t], \mathbf{v}_t\rangle| \le G$. Now we are left to show a bounded central moment:

$$\mathbb{E}[|g_t - \mathbb{E}[g_t]|^{\mathfrak{p}}] = \mathbb{E}[|\langle \mathbf{g}_t - \mathbb{E}[\mathbf{g}_t], \mathbf{v}_t\rangle + \langle \mathbb{E}[\mathbf{g}_t], \mathbf{v}_t\rangle - \mathbb{E}[\langle \mathbf{g}_t, \mathbf{v}_t\rangle]|^{\mathfrak{p}}]$$

$$\le \mathbb{E}[|\langle \mathbf{g}_t - \mathbb{E}[\mathbf{g}_t], \mathbf{v}_t\rangle|^{\mathfrak{p}}] + |\langle \mathbb{E}[\mathbf{g}_t], \mathbf{v}_t\rangle|^{\mathfrak{p}} + |\mathbb{E}[\langle \mathbf{g}_t, \mathbf{v}_t\rangle]|^{\mathfrak{p}}$$

$$\le \mathbb{E}[\|\mathbf{g}_t - \mathbb{E}[\mathbf{g}_t]\|^{\mathfrak{p}}] + 2\|\mathbb{E}[\mathbf{g}_t]\|^{\mathfrak{p}}$$

$$\le \hat{\sigma}^{\mathfrak{p}} + 2G^{\mathfrak{p}} = \sigma^{\mathfrak{p}}$$

Then by Theorem 4, when Algorithm 2 is run on $g_t = \langle \mathbf{g}_t, \mathbf{v}_t\rangle$, we obtain with probability at least $1 - \delta$

$$R_T^{1d}(\|\mathbf{u}\|) \le \epsilon \left(3 + \frac{8\tau\sqrt{2/T}}{\tau + H} \log\left(\frac{224}{\delta}\left[\log\left(1 + \sqrt{T}2^{T+5/2}\right) + 2\right]^2\right)\right)$$

$$+ \|\mathbf{u}\|\frac{3A}{2}\sqrt{N \max\left(0, \tau^2(3 + 2\log\frac{2}{\delta}) + \sum_{t=1}^{T} |\nabla\phi(w_t)|^2 - |\nabla\psi_t(w_t)|^2\right)}$$

$$+ \|\mathbf{u}\|T^{1/\mathfrak{p}}(\sigma^{\mathfrak{p}} + G^{\mathfrak{p}})^{1/\mathfrak{p}}\left[2BN + 2\sqrt{\log\frac{8}{\delta}} + 4\log\frac{8}{\delta}\right.$$

$$+ 4\sqrt{2\log\left(\frac{32}{\delta}\left[\log\left(2^{T+1}\right) + 2\right]^2\right)}\left(\frac{1}{\sqrt{T}}(A^2 + 2B)N + \sqrt{\log\left(\frac{T\|\mathbf{u}\|c_1^2}{\epsilon^2} + 1\right)} + 1\right)$$

$$\left. + \frac{c_2 \log T}{\tau}\left(2(A^2 + B)N + 3\max\left(0, \log T \log\left(\frac{3\|\mathbf{u}\|}{\alpha_2}\right)\right) + 4\right)\right] \tag{20}$$

In terms of $R_T^{nd}(\mathbf{u}/\|\mathbf{u}\|)$, Lemma 16 implies that with probability at least $1 - \delta/2$,

$$R_T^{nd}(\mathbf{u}/\|\mathbf{u}\|) \le T^{1/\mathfrak{p}}(\hat{\sigma}^{\mathfrak{p}} + G^{\mathfrak{p}})^{1/\mathfrak{p}}\left(\frac{9}{2} + \left(\frac{3}{2} + \frac{20}{3}\log\frac{4}{\delta}\right) + 15\sqrt{\log\frac{320}{\delta}}\right.$$

$$+ 184 \log\left(\frac{896}{\delta}\left[\log\left(2 + \frac{16}{\tau}\right) + 1\right]^2\right)\right) \tag{21}$$

Finally, replace $\delta$ as $\delta/2$ in equation (20), then combining with equation (21), we have the regret guarantee with probability at least $1 - \delta$

$$R_T(\mathbf{u}) \le R_T^{1d}(\|\mathbf{u}\|) + \|\mathbf{u}\| R_T^{nd}(\mathbf{u}/\|\mathbf{u}\|)$$

$$\le \epsilon\left(3 + \frac{8\tau\sqrt{2/T}}{\tau + H}\log\left(\frac{448}{\delta}\left[\log\left(1 + \sqrt{T}2^{T+5/2}\right) + 2\right]^2\right)\right)$$

$$+ \|\mathbf{u}\|\frac{3A}{2}\sqrt{N\max\left(0, \tau^2(3 + 2\log\frac{4}{\delta}) + \sum_{t=1}^{T}|\nabla\phi(w_t)|^2 - |\nabla\psi_t(w_t)|^2\right)}$$

$$+ \|\mathbf{u}\|T^{1/\mathfrak{p}}(\sigma^{\mathfrak{p}} + G^{\mathfrak{p}})^{1/\mathfrak{p}}\left[2BN + 2\sqrt{\log\frac{16}{\delta}} + 4\log\frac{16}{\delta}\right.$$

$$+ 4\sqrt{2\log\left(\frac{64}{\delta}\left[\log\left(2^{T+1}\right) + 2\right]^2\right)}\left(\frac{1}{\sqrt{T}}(A^2 + 2B)N + \sqrt{\log\left(\frac{T\|\mathbf{u}\|c_1^2}{\epsilon^2} + 1\right)} + 1\right)$$

$$+ \frac{c_2\log T}{\tau}\left(2(A^2 + B)N + 3\max\left(0, \log T\log\left(\frac{3\|\mathbf{u}\|}{\alpha_2}\right)\right) + 4\right)\right]$$

$$+ \|\mathbf{u}\|T^{1/\mathfrak{p}}(\hat{\sigma}^{\mathfrak{p}} + G^{\mathfrak{p}})^{1/\mathfrak{p}}\left[\frac{9}{2} + \left(\frac{3}{2} + \frac{20}{3}\log\frac{4}{\delta}\right)\right.$$

$$\left. + 15\sqrt{\log\frac{320}{\delta}} + 184\log\left(\frac{896}{\delta}\left[\log\left(2 + \frac{16}{\tau}\right) + 1\right]^2\right)\right]$$

$\square$

## E   Technical concentration bounds

In this section we collect some technical results on concentration martingale difference sequences. These results are not new, and the proofs are for the most part exercises in known techniques (e.g. Howard et al. [2021], Balsubramani [2014]), but we cannot find simple explicit statements of exactly the forms we require in the literature, so we provide them here and include proofs for completeness. Our approach follows the martingale mixture method used by Balsubramani [2014]. We make no effort to achieve optimal constants, in several cases explicitly choosing weaker bounds to make the final numbers less complicated.

To start, we recall the notion of a sub-exponential martingale difference sequence (MDS) as follows:

**Definition 1.** *Suppose $\{X_t\}$ is a sequence of random variables adapted to a filtration $\mathcal{F}_t$ such that $\{X_t, \mathcal{F}_t\}$ is a martingale difference sequence. Further, suppose $\{\sigma_t, b_t\}$ are random variables such that $\sigma_t, b_t$ are both $\mathcal{F}_{t-1}$-measurable for all $t$. Then, $\{X_t, \mathcal{F}_t\}$ is $\{\sigma_t, b_t\}$ sub-exponential if*

$$\mathbb{E}[\exp(\lambda X_t)|\mathcal{F}_{t-1}] \le \exp(\lambda^2\sigma_t^2/2)$$

*almost everywhere for all $\mathcal{F}_{t-1}$-measurable $\lambda$ satisfying $\lambda < 1/b_t$.*

We have the following useful way to obtain sub-exponential tails:

**Proposition 17.** *Suppose $\{X_t, \mathcal{F}_t\}$ is a MDS such that $\mathbb{E}[X_t^2|\mathcal{F}_t] \le \sigma_t^2$ and $|X_t| \le b_t$ almost everywhere for all $t$ for some sequence of random variable $\{\sigma_t, b_t\}$ such that $\sigma_t, b_t$ is $\mathcal{F}_{t-1}$-measurable. Then $X_t$ is $(\sigma_t, 2b_t)$ sub-exponential.*

The main results of this section are the following two Theorems. First for scalar random variables we have the following one-sided concentration bound:

**Theorem 18.** *Suppose $\{X_t, \mathcal{F}_t\}$ is a $(\sigma_t, b_t)$ sub-exponential martingale difference sequence. Let $\nu$ be an arbitrary constant. Then with probability at least $1 - \delta$, for all $t$ it holds that:*

$$\sum_{i=1}^{t} X_i \leq 2\sqrt{\sum_{i=1}^{t} \sigma_i^2 \log\left(\frac{4}{\delta}\left[\log\left(\left[\sqrt{\sum_{i=1}^{t} \sigma_i^2/2\nu^2}\right]_1\right) + 2\right]^2\right)}$$

$$+ 8\max(\nu, \max_{i \leq t} b_i) \log\left(\frac{28}{\delta}\left[\log\left(\frac{\max(\nu, \max_{i \leq t} b_i)}{\nu}\right) + 2\right]^2\right)$$

*where $[x]_1 = \max(1, x)$.*

Next, for vector valued random variables, we have the following bound:

**Theorem 19.** *Suppose that $\{X_t, \mathcal{F}_t\}$ is a vector-valued martingale difference sequence such that $\mathbb{E}[\|X_t\|^2 | \mathcal{F}_{t-1}] \leq \sigma_t^2$ and $\|X_t\| \leq b_t$ almost everywhere for some sequence $\{\sigma_t, b_t\}$ such that $\sigma_t, b_t$ is $\mathcal{F}_{t-1}$-measurable. Let $\nu \geq 0$ be an arbitrary constant. Then with probability at least $1 - \delta$, for all $t$ we have:*

$$\left\|\sum_{i=1}^{t} X_i\right\| \leq 5\sqrt{\sum_{i=1}^{t} \sigma_i^2 \log\left(\frac{16}{\delta}\left[\log\left(\left[\sqrt{\sum_{i=1}^{t} \sigma_i^2/\nu^2}\right]_1\right) + 2\right]^2\right)}$$

$$+ 23\max(\nu, \max_{i \leq t} b_i) \log\left(\frac{224}{\delta}\left[\log\left(\frac{2\max(\nu, \max_{i \leq t} b_i)}{\nu}\right) + 2\right]^2\right)$$

*where $[x]_1 = \max(1, x)$.*

### E.1 Time-Uniform Concentration of sums of sub-exponential MDS

In this section, we will prove Theorem 18.

**Theorem 18.** *Suppose $\{X_t, \mathcal{F}_t\}$ is a $(\sigma_t, b_t)$ sub-exponential martingale difference sequence. Let $\nu$ be an arbitrary constant. Then with probability at least $1 - \delta$, for all $t$ it holds that:*

$$\sum_{i=1}^{t} X_i \leq 2\sqrt{\sum_{i=1}^{t} \sigma_i^2 \log\left(\frac{4}{\delta}\left[\log\left(\left[\sqrt{\sum_{i=1}^{t} \sigma_i^2/2\nu^2}\right]_1\right) + 2\right]^2\right)}$$

$$+ 8\max(\nu, \max_{i \leq t} b_i) \log\left(\frac{28}{\delta}\left[\log\left(\frac{\max(\nu, \max_{i \leq t} b_i)}{\nu}\right) + 2\right]^2\right)$$

*where $[x]_1 = \max(1, x)$.*

*Proof.* First, observe that by replacing $b_t$ with $\max_{i \leq t} b_i$, we may assume $b_t \geq b_{t-1}$ for all $t$ with probability 1. Notice that after this operation, $X_t$ is still $(\sigma_t, b_t)$ sub-exponential. Under this assumption, it suffices to prove the result with $b_t$ in place of $\max_{i \leq t} b_i$.

Define $M_t = \sum_{i=1}^{t} X_i$ and let $\pi(\eta)$ be a to-be-specified probability density function on $\mathbb{R}$. Define:

$$Z_t = \int_0^{1/\max(\nu, b_t)} \pi(\eta) \exp\left[\eta M_t - \frac{\eta^2 \sum_{i=1}^{t} \sigma_i^2}{2}\right] d\eta$$

Notice that $Z_0 \leq \int_0^\infty \pi(\eta) \, d\eta = 1$. We claim that $Z_t$ is itself a supermartingale adapted to the same filtration $\mathcal{F}_t$:

$$\mathbb{E}[Z_t | \mathcal{F}_{t-1}] = \mathbb{E}\left[\int_0^{1/\max(\nu, b_t)} \pi(\eta) \exp\left[\eta M_t - \frac{\eta^2 \sum_{i=1}^{t} \sigma_i^2}{2}\right] d\eta \Big| \mathcal{F}_{t-1}\right]$$

$$= \mathbb{E}\left[\int_0^{1/\max(\nu,b_t)} \pi(\eta) \exp\left[\eta M_{t-1} - \frac{\eta^2 \sum_{i=1}^{t-1} \sigma_i^2}{2}\right] \exp\left[\eta X_t - \frac{\eta^2 \sigma_t^2}{2}\right] d\eta \mid \mathcal{F}_{t-1}\right]$$

$$= \int_0^{1/\max(\nu,b_t)} \pi(\eta) \exp\left[\eta M_{t-1} - \frac{\eta^2 \sum_{i=1}^{t-1} \sigma_i^2}{2}\right] \mathbb{E}\left[\exp\left[\eta X_t - \frac{\eta^2 \sigma_t^2}{2}\right] \mid \mathcal{F}_{t-1}\right] d\eta$$

Use the sub-exponentiality of $X_t$:

$$\leq \int_0^{1/\max(\nu,b_t)} \pi(\eta) \exp\left[\eta M_{t-1} - \frac{\eta^2 \sum_{i=1}^{t-1} \sigma_i^2}{2}\right] d\eta$$

Use $b_t \geq b_{t-1}$:

$$\leq \int_0^{1/\max(\nu,b_{t-1})} \pi(\eta) \exp\left[\eta M_{t-1} - \frac{\eta^2 \sum_{i=1}^{t-1} \sigma_i^2}{2}\right] d\eta$$

$$= Z_{t-1}$$

Therefore, by Ville's maximal inequality [Ville, 1939], we have that for all $\delta > 0$:

$$P\left[\sup_t Z_t \geq 1/\delta\right] \leq \delta Z_0 = \delta$$

Put another way, with probability at least $1 - \delta$, $Z_t \leq 1/\delta$ for all $t$.

Now, let us define $\pi(\eta)$. With the benefit of foresight, we choose a density on $[0, 1/\nu]$:

$$\pi(\eta) = \frac{1}{\eta(\log(1/\eta\nu) + 2)^2} \tag{22}$$

We have

$$\pi(\eta) = \frac{d}{d\eta} \frac{1}{\log(1/\eta\nu) + 2} \tag{23}$$

$$\frac{d\pi(\eta)}{d\eta} = -\frac{\log(1/\eta\nu)}{\eta^2 \left(\log(1/\eta\nu) + 2\right)^3} \tag{24}$$

so that $\int_0^{1/\nu} \pi(\eta)\, d\eta = 1$ and $\pi(\eta)$ is decreasing on $[0, 1/\nu]$.

Next, for any given $\eta_\star \in [0, 1/\nu]$, for all $K \geq 1$, for all $\eta \in [\eta_\star/K, \eta_\star]$, we have:

$$\int_{\eta_\star/K}^{\eta_\star} \pi(\eta)\, d\eta \geq \pi(\eta_\star)\frac{K-1}{K}\eta_\star$$

$$\geq \frac{(K-1)}{K(\log(1/\eta_\star\nu) + 2)^2}$$

Further, for all $\eta \in [\eta_\star/K, \eta_\star]$, if $M_t \geq 0$ we also have:

$$\eta M_t - \frac{\eta^2 \sum_{i=0}^{t-1} \sigma_i^2}{2} \geq \frac{\eta_\star M_t}{K} - \frac{\eta_\star^2 \sum_{i=0}^{t-1} \sigma_i^2}{2}$$

and otherwise of course we have $M_t \leq 0$.

Combining these observations, we have that with probability at least $1 - \delta$, for all $\mathcal{F}_{t-1}$-measurable $\eta_\star$ satisfying $\eta_\star \leq 1/\max(\nu, b_t)$ and all $K \geq 1$, either $M_t \leq 0$ or

$$\frac{(K-1)}{K(\log(1/\eta_\star\nu) + 2)^2} \exp\left(\frac{\eta_\star M_t}{K} - \frac{\eta_\star^2 \sum_{i=1}^t \sigma_i^2}{2}\right) \leq \int_0^{1/b_t} \pi(\eta) \exp\left[\eta M_t - \frac{\eta^2 \sum_{i=1}^t \sigma_i^2}{2}\right] d\eta \leq \delta$$

Now, rearranging this identity implies:

$$M_t \leq \log\left(\frac{K}{(K-1)\delta}\left[\log\left(\frac{1}{\eta_\star \nu}\right)+2\right]^2\right)\frac{K}{\eta_\star}+\frac{K\eta_\star \sum_{i=1}^t \sigma_i^2}{2} \tag{25}$$

So that overall we may discard the $M_t \leq 0$ case as it is strictly weaker than the above.

Now, again with the benefit of foresight, let us select

$$\eta_\star = \min\left(\frac{1}{\max(\nu, b_t)}, \sqrt{\frac{2\log\left[\frac{K}{(K-1)\delta}\left[\log\left(\left[\sqrt{\sum_{i=1}^t \sigma_i^2/2\nu^2}\right]_1\right)+2\right]^2\right]}{\sum_{i=1}^t \sigma_i^2}}\right)$$

where $[x]_1 = \max(1, x)$. Notice that $\eta_\star$ is $\mathcal{F}_{t-1}$-measurable since $b_t$ and $\sigma_1, \ldots, \sigma_t$ are $\mathcal{F}_{t-1}$-measurable, and $\eta_\star \in [0, 1/\nu]$ with probability 1.

Now, to analyze the expression (25), we will consider both cases of the above minimum. First, let us assume

$$1/\max(\nu, b_t) \geq \sqrt{\frac{2\log\left[\frac{K}{(K-1)\delta}\left[\log\left(\left[\sqrt{\sum_{i=1}^t \sigma_i^2/2\nu^2}\right]_1\right)+2\right]^2\right]}{\sum_{i=1}^t \sigma_i^2}}$$

Then we can bound $\eta_\star$:

$$\eta_\star \geq \sqrt{\frac{2\log\left[\frac{K}{(K-1)\delta}[2]^2\right]}{\sum_{i=1}^t \sigma_i^2}}$$

$$\geq \sqrt{\frac{2}{\sum_{i=1}^t \sigma_i^2}}$$

Therefore:

$$\log\left(\frac{K}{(K-1)\delta}\left[\log\left(\frac{1}{\eta_\star \nu}\right)+2\right]^2\right) \leq \log\left(\frac{K}{(K-1)\delta}\left[\log\left(\left[\sqrt{\sum_{i=0}^{t-1}\sigma_i^2/2\nu^2}\right]_1\right)+2\right]^2\right)$$

from which we conclude:

$$M_t \leq K\sqrt{2\sum_{i=1}^t \sigma_i^2 \log\left(\frac{K}{(K-1)\delta}\left[\log\left(\left[\sqrt{\sum_{i=1}^t \sigma_i^2/2\nu^2}\right]_1\right)+2\right]^2\right)}$$

Now, on the other hand let us suppose that

$$1/\max(\nu, b_t) < \sqrt{\frac{2\log\left[\frac{K}{(K-1)\delta}\left[\log\left(\left[\sqrt{\sum_{i=1}^t \sigma_i^2/2\nu^2}\right]_1\right)+2\right]^2\right]}{\sum_{i=1}^t \sigma_i^2}}$$

This implies:

$$\sum_{i=1}^t \sigma_i^2 < 2\min(\nu^2, b_t^2)\log\left[\frac{K}{(K-1)\delta}\left[\log\left(\left[\sqrt{\sum_{i=1}^t \sigma_i^2/2\nu^2}\right]_1\right)+2\right]^2\right]$$

Therefore, from Lemma 25 we have:

$$\sum_{i=1}^{t} \sigma_i^2 \le 8 \max(\nu^2, b_t^2) \log \left[ \frac{4\sqrt{K}}{\sqrt{(K-1)\delta}} \log \left( e + 16 \frac{\max(\nu^2, b_t^2)}{\nu^2} \right) \right]$$

$$\le 8 \max(\nu^2, b_t^2) \log \left[ \frac{4\sqrt{K}}{\sqrt{(K-1)\delta}} \log \left( 20 \frac{\max(\nu^2, b_t^2)}{\nu^2} \right) \right]$$

In this case, we will have $\eta_\star = 1/\max(\nu, b_t)$ and so obtain:

$$M_t \le K \max(\nu, b_t) \log \left( \frac{K}{(K-1)\delta} \left[ \log \left( \frac{\max(\nu, b_t)}{\nu} \right) + 2 \right]^2 \right)$$

$$+ 4 \max(\nu, b_t) \log \left[ \frac{4\sqrt{K}}{\sqrt{(K-1)\delta}} \log \left( 20 \frac{\max(\nu^2, b_t^2)}{\nu^2} \right) \right]$$

$$\le K \max(\nu, b_t) \log \left( \frac{K}{(K-1)\delta} \left[ \log \left( \frac{\max(\nu, b_t)}{\nu} \right) + 2 \right]^2 \right)$$

$$+ 4 \max(\nu, b_t) \log \left[ \frac{8\sqrt{K}}{\sqrt{(K-1)\delta}} \log \left( \frac{5 \max(\nu, b_{t-1})}{\nu} \right) \right]$$

$$\le K \max(\nu, b_t) \log \left( \frac{K}{(K-1)\delta} \left[ \log \left( \frac{\max(\nu, b_t)}{\nu} \right) + 2 \right]^2 \right)$$

$$+ 4 \max(\nu, b_t) \log \left[ \frac{8\sqrt{K}}{\sqrt{(K-1)\delta}} \left[ \log \left( \frac{\max(\nu, b_t)}{\nu} \right) + 2 \right] \right]$$

$$\le 5K \max(\nu, b_t) \log \left( \frac{8K}{(K-1)\delta} \left[ \log \left( \frac{\max(\nu, b_t)}{\nu} \right) + 2 \right]^2 \right)$$

Finally, let us set $K = \sqrt{2}$ to obtain:

$$M_t \le K \sqrt{ 2 \sum_{i=1}^{t} \sigma_i^2 \log \left( \frac{K}{(K-1)\delta} \left[ \log \left( \left[ \sqrt{\sum_{i=1}^{t} \sigma_i^2 / 2\nu^2} \right]_1 \right) + 2 \right]^2 \right) }$$

$$+ 5K \max(\nu, b_t) \log \left( \frac{8K}{(K-1)\delta} \left[ \log \left( \frac{\max(\nu, b_{t-1})}{\nu} \right) + 2 \right]^2 \right)$$

$$\le 2 \sqrt{ \sum_{i=1}^{t} \sigma_i^2 \log \left( \frac{4}{\delta} \left[ \log \left( \left[ \sqrt{\sum_{i=1}^{t} \sigma_i^2 / 2\nu^2} \right]_1 \right) + 2 \right]^2 \right) }$$

$$+ 8 \max(\nu, b_t) \log \left( \frac{28}{\delta} \left[ \log \left( \frac{\max(\nu, b_t)}{\nu} \right) + 2 \right]^2 \right)$$

$$\square$$

## E.2 Bounds on Sums of Squares

It is also often useful to bound sums of the form $\sum_{i=1}^{t} Z_i^2$ for some sequence $Z_i$. Here we collect a useful bound:

**Theorem 20.** *Suppose $\{Z_i\}$ is a sequence of random variables adapted to a filtration $\{\mathcal{F}_t\}$. Further, suppose $\mathbb{E}[Z_i^2] \le \sigma_i^2$ and $|Z_i| \le b_i$ for all $i$ with probability 1 for some $\sigma_i$ and $b_i$ for a sequence*

$\{\sigma_i, b_i\}$ *such that $\sigma_i$ and $b_i$ are $\mathcal{F}_{i-1}$-measurable. Then for any $\nu > 0$, with probability at least $1 - \delta$ for all $t$:*

$$\sum_{i=1}^{t} Z_i^2 \le 3 \sum_{i=1}^{t} \sigma_i^2 \log \left( \frac{4}{\delta} \left[ \log \left( \left[ \sqrt{\sum_{i=1}^{t} \sigma_i^2/\nu^2} \right]_1 \right) + 2 \right]^2 \right)$$

$$+ 20 \max(\nu^2, \max_{i \le t} b_i^2) \log \left( \frac{112}{\delta} \left[ \log \left( \frac{2 \max(\nu, \max_{i \le t} b_i)}{\nu} \right) + 1 \right]^2 \right)$$

*Proof.* Define $X_t = Z_t^2 - \mathbb{E}[Z_t^2|\mathcal{F}_{t-1}]$ so that $\{X_t, \mathcal{F}_t\}$ is a MDS. Further, notice that $|X_t| \le b_t^2$ for all $t$ with probability 1 and

$$\mathbb{E}[X_t^2|\mathcal{F}_{t-1}] \le b_t^2 \mathbb{E}[|Z_t^2 - \mathbb{E}[Z_t^2|\mathcal{F}_{t-1}]| \mid \mathcal{F}_{t-1}]$$
$$\le 2b_t^2 \sigma_t^2$$

Therefore, by Proposition 17, $X_t$ is $(\sqrt{2b_t^2 \sigma_t^2}, 2b_t^2)$ sub-exponential.

Thus, by our time-uniform concentration bound (Theorem 18), for any $\nu > 0$, with probability at least $1 - \delta$ we have:

$$\sum_{i=1}^{t} X_i \le 2 \sqrt{2 \sum_{i=1}^{t} \sigma_i^2 b_i^2 \log \left( \frac{4}{\delta} \left[ \log \left( \left[ \sqrt{\sum_{i=1}^{t} \sigma_i^2 b_i^2/\nu^4} \right]_1 \right) + 2 \right]^2 \right)}$$

$$+ 8 \max(\nu^2, 2 \max_{i \le t} b_t^2) \log \left( \frac{28}{\delta} \left[ \log \left( \frac{\max(\nu^2, 2 \max_{i \le t} b_i^2)}{\nu^2} \right) + 2 \right]^2 \right)$$

$$\le 2 \sqrt{2 \max_{i \le t} b_i^2 \sum_{i=1}^{t} \sigma_i^2 \log \left( \frac{4}{\delta} \left[ \log \left( \left[ \sqrt{\sum_{i=1}^{t} \sigma_i^2 b_i^2/\nu^4} \right]_1 \right) + 2 \right]^2 \right)}$$

$$+ 8 \max(\nu^2, 2 \max_{i \le t} b_i^2) \log \left( \frac{28}{\delta} \left[ \log \left( \frac{\max(\nu^2, 2 \max_{i \le t} b_i^2)}{\nu^2} \right) + 2 \right]^2 \right)$$

Now, by Young inequality:

$$2 \max_{i \le t} b_i^2 \sum_{i=1}^{t} \sigma_i^2 \log \left( \frac{4}{\delta} \left[ \log \left( \left[ \sqrt{\sum_{i=1}^{t} \sigma_i^2 b_i^2/\nu^4} \right]_1 \right) + 2 \right]^2 \right)$$

$$\le \left( \max_{i \le t} b_i^2 \log \left( \frac{28}{\delta} \left[ \log \left( \frac{\max(\nu^2, 2 \max_{i \le t} b_i^2)}{\nu^2} \right) + 2 \right]^2 \right) \right)^2$$

$$+ \left( \sum_{i=1}^{t} \sigma_i^2 \frac{\log \left( \frac{4}{\delta} \left[ \log \left( \left[ \sqrt{\sum_{i=1}^{t} \sigma_i^2 b_i^2/\nu^4} \right]_1 \right) + 2 \right]^2 \right)}{\log \left( \frac{28}{\delta} \left[ \log \left( \frac{\max(\nu^2, 2 \max_{i \le t} b_i^2)}{\nu^2} \right) + 2 \right]^2 \right)} \right)^2$$

To simplify this expression, we consider the following identity:

$$\frac{\log \left( \frac{4}{\delta} \left[ \log \left( \left[ \sqrt{\sum_{i=1}^{t} \sigma_i^2 b_i^2/\nu^4} \right]_1 \right) + 2 \right]^2 \right)}{\log \left( \frac{28}{\delta} \left[ \log \left( \frac{\max(\nu^2, 2 \max_{i \le t} b_i^2)}{\nu^2} \right) + 2 \right]^2 \right)} \le \frac{\log \left( \frac{4}{\delta} \left[ \log \left( \left[ \sqrt{\sum_{i=1}^{t} \sigma_i^2/\nu^2} \sqrt{\max_{i \le t} b_i^2/\nu^2} \right]_1 \right) + 2 \right]^2 \right)}{\log \left( \frac{28}{\delta} \left[ \log \left( \frac{\max(\nu^2, 2 \max_{i \le t} b_i^2)}{\nu^2} \right) + 2 \right]^2 \right)}$$

$$\leq \frac{\log\left(\frac{4}{\delta}\left[\log\left(\left[\sqrt{\sum_{i=1}^{t}\sigma_i^2/\nu^2}\right]_1\right)+\frac{1}{2}\log\left(\frac{\max(\nu^2,2\max_{i\leq t}b_i^2)}{\nu^2}\right)+2\right]^2\right)}{\log\left(\frac{28}{\delta}\left[\log\left(\frac{\max(\nu^2,2\max_{i\leq t}b_i^2)}{\nu^2}\right)+2\right]^2\right)}$$

Now, consider two cases, either

$$\log\left(\left[\sqrt{\sum_{i=1}^{t}\sigma_i^2/\nu^2}\right]_1\right)\leq\frac{1}{2}\log\left(\frac{\max(\nu^2,2\max_{i\leq t}b_i^2)}{\nu^2}\right)$$

or not. In the former case, we have:

$$\frac{\log\left(\frac{4}{\delta}\left[\log\left(\left[\sqrt{\sum_{i=1}^{t}\sigma_i^2 b_i^2/\nu^4}\right]_1\right)+2\right]^2\right)}{\log\left(\frac{28}{\delta}\left[\log\left(\frac{2\max(\nu^2,2\max_{i\leq t}b_i^2)}{\nu^2}\right)+2\right]^2\right)}\leq\frac{\log\left(\frac{4}{\delta}\left[\log\left(\left[\sqrt{\sum_{i=1}^{t}\sigma_i^2/\nu^2}\right]_1\right)+\frac{1}{2}\log\left(\frac{\max(\nu^2,2\max_{i\leq t}b_i^2)}{\nu^2}\right)+2\right]^2\right)}{\log\left(\frac{28}{\delta}\left[\log\left(\frac{\max(\nu^2,2\max_{i\leq t}b_i^2)}{\nu^2}\right)+2\right]^2\right)}$$

$$\leq\frac{\log\left(\frac{4}{\delta}\left[\log\left(\frac{\max(\nu^2,2\max_{i\leq t}b_i^2)}{\nu^2}\right)+2\right]^2\right)}{\log\left(\frac{28}{\delta}\left[\log\left(\frac{\max(\nu^2,2\max_{i\leq t}b_i^2)}{\nu^2}\right)+2\right]^2\right)}$$

$$\leq 1$$

$$\leq\log\left(\frac{4}{\delta}\left[\log\left(\left[\sqrt{\sum_{i=1}^{t}\sigma_i^2/\nu^2}\right]_1\right)+2\right]^2\right)$$

While in the latter case,

$$\frac{\log\left(\frac{4}{\delta}\left[\log\left(\left[\sqrt{\sum_{i=1}^{t}\sigma_i^2 b_i^2/\nu^4}\right]_1\right)+2\right]^2\right)}{\log\left(\frac{28}{\delta}\left[\log\left(\frac{\max(\nu^2,2\max_{i\leq t}b_i^2)}{\nu^2}\right)+2\right]^2\right)}\leq\frac{\log\left(\frac{4}{\delta}\left[2\log\left(\sqrt{\sum_{i=1}^{t}\sigma_i^2/\nu^2}\right)+2\right]^2\right)}{\log\left(\frac{28}{\delta}\left[\log\left(\frac{\max(\nu^2,2\max_{i\leq t}b_i^2)}{\nu^2}\right)+2\right]^2\right)}$$

$$\leq\log\left(\frac{4}{\delta}\left[\log\left(\left[\sqrt{\sum_{i=1}^{t}\sigma_i^2/\nu^2}\right]_1\right)+2\right]^2\right)$$

So in both cases, we have

$$\frac{\log\left(\frac{4}{\delta}\left[\log\left(\left[\sqrt{\sum_{i=1}^{t}\sigma_i^2 b_i^2/\nu^4}\right]_1\right)+2\right]^2\right)}{\log\left(\frac{28}{\delta}\left[\log\left(\frac{\max(\nu^2,2\max_{i\leq t}b_i^2)}{\nu^2}\right)+2\right]^2\right)}\leq\log\left(\frac{4}{\delta}\left[\log\left(\left[\sqrt{\sum_{i=1}^{t}\sigma_i^2/\nu^2}\right]_1\right)+2\right]^2\right)$$

Therefore,

$$2\max_{i\leq t}b_i^2\sum_{i=1}^{t}\sigma_i^2\log\left(\frac{4}{\delta}\left[\log\left(\left[\sqrt{\sum_{i=1}^{t}\sigma_i^2 b_i^2/\nu^4}\right]_1\right)+2\right]^2\right)$$

$$\leq\left(\max_{i\leq t}b_i^2\log\left(\frac{28}{\delta}\left[\log\left(\frac{\max(\nu^2,2\max_{i\leq t}b_i^2)}{\nu^2}\right)+2\right]^2\right)\right)^2$$

$$+\left(\sum_{i=1}^{t}\sigma_i^2\log\left(\frac{4}{\delta}\left[\log\left(\left[\sqrt{\sum_{i=1}^{t}\sigma_i^2/\nu^2}\right]_1\right)+2\right]^2\right)\right)^2$$

Combining this with the identity $\sqrt{a+b} \leq \sqrt{a} + \sqrt{b}$, we have:

$$\sum_{i=1}^{t} X_i \leq 2 \sqrt{ 2 \max_{i \leq t} b_i^2 \sum_{i=1}^{t} \sigma_i^2 \log\left( \frac{4}{\delta} \left[ \log\left( \left[ \sqrt{\sum_{i=1}^{t} \sigma_i^2 b_i^2 / \nu^4} \right]_1 \right) + 2 \right]^2 \right) }$$

$$+ 8 \max(\nu^2, 2 \max_{i \leq t} b_i^2) \log\left( \frac{28}{\delta} \left[ \log\left( \frac{\max(\nu^2, 2\max_{i \leq t} b_i^2)}{\nu^2} \right) + 2 \right]^2 \right)$$

$$\leq 2 \sum_{i=1}^{t} \sigma_i^2 \log\left( \frac{4}{\delta} \left[ \log\left( \left[ \sqrt{\sum_{i=1}^{t} \sigma_i^2 / \nu^2} \right]_1 \right) + 2 \right]^2 \right)$$

$$+ 10 \max(\nu^2, 2 \max_{i \leq t} b_i^2) \log\left( \frac{28}{\delta} \left[ \log\left( \frac{\max(\nu^2, 2\max_{i \leq t} b_i^2)}{\nu^2} \right) + 2 \right]^2 \right)$$

$$\leq 2 \sum_{i=1}^{t} \sigma_i^2 \log\left( \frac{4}{\delta} \left[ \log\left( \left[ \sqrt{\sum_{i=1}^{t} \sigma_i^2 / \nu^2} \right]_1 \right) + 2 \right]^2 \right)$$

$$+ 20 \max(\nu^2, \max_{i \leq t} b_i^2) \log\left( \frac{112}{\delta} \left[ \log\left( \frac{2\max(\nu, \max_{i \leq t} b_i)}{\nu} \right) + 1 \right]^2 \right)$$

Finally, observe that

$$\sum_{i=1}^{t} Z_i^2 = \sum_{i=1}^{t} X_i + \mathbb{E}[Z_i^2 | \mathcal{F}_{t-1}]$$

$$\leq \sum_{i=1}^{t} \sigma_i^2 + \sum_{i=1}^{t} X_i$$

$$\leq 3 \sum_{i=1}^{t} \sigma_i^2 \log\left( \frac{4}{\delta} \left[ \log\left( \left[ \sqrt{\sum_{i=1}^{t} \sigma_i^2 / \nu^2} \right]_1 \right) + 1 \right]^2 \right)$$

$$+ 20 \max(\nu^2, \max_{i \leq t} b_i^2) \log\left( \frac{112}{\delta} \left[ \log\left( \frac{2\max(\nu, \max_{i \leq t} b_i)}{\nu} \right) + 1 \right]^2 \right)$$

$\square$

### E.3  From scalar to vector (Hilbert space) concentration

In this section we extend our results to concentration of norm of vectors in Hilbert space. The technique follows that of Cutkosky and Mehta [2021], which makes use of a particular scalar sequence associated to any vector sequence described by Cutkosky [2018]. Given any sequence $X_1, X_2, \ldots$ of vectors, define a sequence $s_1, s_2, \ldots$ of scalars as follows:

1. $s_0 = 0$.
2. If $\sum_{i=1}^{t-1} X_i \neq 0$, set

$$s_t = \text{sign}\left( \sum_{i=1}^{t-1} s_i \right) \frac{\left\langle \sum_{i=1}^{t-1} X_i, X_t \right\rangle}{\left\| \sum_{i=1}^{t-1} X_i \right\|}$$

where we define $\text{sign}(z) = 1$ if $z \geq 0$ and $\text{sign}(z) = -1$ otherwise.
3. If $\sum_{i=1}^{t-1} X_i = 0$, set $s_t = 0$.

Clearly if $\{X_t\}$ is a random sequence adapted to the filtration $\mathcal{F}_t$, then so is $s_t$. Now, these $s_t$ have the following interesting property:

**Lemma 21** (Cutkosky and Mehta [2021], Lemma 10). *For all $t$, we have $|s_t| \leq \|X_t\|$ and*

$$\left\| \sum_{i=1}^{t} X_i \right\| \leq \left| \sum_{i=1}^{t} s_i \right| + \sqrt{\max_{i \leq t} \|X_i\|^2 + \sum_{i=1}^{t} \|X_i\|^2}$$

Now, we need to use this result. The key is that if $X_t$ is a MDS, then the $s_t$ will be also. Thus, we can bound the sum of the $X_t$ using the sum of the $s_t$, which can in turn be bounded by *scalar* martingale concentration bounds. Let us instantiate this using our previous bounds to obtain:

**Theorem 19.** *Suppose that $\{X_t, \mathcal{F}_t\}$ is a vector-valued martingale difference sequence such that $\mathbb{E}[\|X_t\|^2 | \mathcal{F}_{t-1}] \leq \sigma_t^2$ and $\|X_t\| \leq b_t$ almost everywhere for some sequence $\{\sigma_t, b_t\}$ such that $\sigma_t, b_t$ is $\mathcal{F}_{t-1}$-measurable. Let $\nu \geq 0$ be an arbitrary constant. Then with probability at least $1 - \delta$, for all $t$ we have:*

$$\left\| \sum_{i=1}^{t} X_i \right\| \leq 5 \sqrt{\sum_{i=1}^{t} \sigma_i^2 \log \left( \frac{16}{\delta} \left[ \log \left( \left[ \sqrt{\sum_{i=1}^{t} \sigma_i^2/\nu^2} \right]_1 \right) + 2 \right]^2 \right)}$$

$$+ 23 \max(\nu, \max_{i \leq t} b_i) \log \left( \frac{224}{\delta} \left[ \log \left( \frac{2\max(\nu, \max_{i \leq t} b_i)}{\nu} \right) + 2 \right]^2 \right)$$

*where $[x]_1 = \max(1, x)$.*

*Proof.* Observe from the construction of the $s_t$ sequence that $\{s_t\}$ is a MDS adapted to $\mathcal{F}_t$, and that $\mathbb{E}[s_t^2 | \mathcal{F}_{t-1}] \leq \sigma_{t-1}^2$ and $|s_t| \leq b_{t-1}$. Therefore $s_t$ is $\sigma_t, 2b_t$ subgaussian. Invoking Theorem 18 (with a union bound for a two-sided inequality), with probability at least $1 - \delta/2$ we have:

$$\left| \sum_{i=1}^{t} s_i \right| \leq 2 \sqrt{\sum_{i=1}^{t} \sigma_i^2 \log \left( \frac{16}{\delta} \left[ \log \left( \left[ \sqrt{\sum_{i=1}^{t} \sigma_i^2/2\nu^2} \right]_1 \right) + 2 \right]^2 \right)}$$

$$+ 16 \max(\nu, \max_{i \leq t} b_i) \log \left( \frac{112}{\delta} \left[ \log \left( \frac{2\max(\nu, \max_{i \leq t} b_i)}{\nu} \right) + 2 \right]^2 \right)$$

Next, observe that $\|X_t\|$ satisfies the conditions of Theorem 20 so that also with probability at least $1 - \delta/2$:

$$\sum_{i=1}^{t} \|X_i\|^2 \leq 3 \sum_{i=1}^{t} \sigma_i^2 \log \left( \frac{8}{\delta} \left[ \log \left( \left[ \sqrt{\sum_{i=1}^{t} \sigma_i^2/\nu^2} \right]_1 \right) + 2 \right]^2 \right)$$

$$+ 20 \max(\nu^2, \max_{i \leq t} b_i^2) \log \left( \frac{224}{\delta} \left[ \log \left( \frac{2\max(\nu, \max_{i \leq t} b_i)}{\nu} \right) + 1 \right]^2 \right)$$

Putting this together with Lemma 21 we have

$$\left\| \sum_{i=1}^{t} X_i \right\| \leq \left| \sum_{i=1}^{t} s_i \right| + \sqrt{\max_{i \leq t-1} \|X_i\|^2 + \sum_{i=1}^{t} \|X_i\|^2}$$

$$\leq \left| \sum_{i=1}^{t} s_i \right| + \sqrt{2 \sum_{i=1}^{t} \|X_i\|^2}$$

$$\leq 2\sqrt{\sum_{i=1}^{t}\sigma_i^2 \log\left(\frac{16}{\delta}\left[\log\left(\left[\sqrt{\sum_{i=1}^{t}\sigma_i^2/2\nu^2}\right]_1\right)+2\right]^2\right)}$$

$$+16\max(\nu,\max_{i\leq t}b_i)\log\left(\frac{112}{\delta}\left[\log\left(\frac{2\max(\nu,\max_{i\leq t}b_i)}{\nu}\right)+2\right]^2\right)$$

$$+\sqrt{6\sum_{i=1}^{t}\sigma_i^2 \log\left(\frac{8}{\delta}\left[\log\left(\left[\sqrt{\sum_{i=1}^{t}\sigma_i^2/\nu^2}\right]_1\right)+2\right]^2\right)}$$

$$+\sqrt{40\max(\nu^2,\max_{i\leq t}b_i^2)\log\left(\frac{224}{\delta}\left[\log\left(\frac{2\max(\nu,\max_{i\leq t}b_i)}{\nu}\right)+1\right]^2\right)}$$

$$\leq 5\sqrt{\sum_{i=1}^{t}\sigma_i^2 \log\left(\frac{16}{\delta}\left[\log\left(\left[\sqrt{\sum_{i=1}^{t}\sigma_i^2/\nu^2}\right]_1\right)+2\right]^2\right)}$$

$$+23\max(\nu,\max_{i\leq t}b_i)\log\left(\frac{224}{\delta}\left[\log\left(\frac{2\max(\nu,\max_{i\leq t}b_i)}{\nu}\right)+2\right]^2\right)$$

$$\square$$

### E.4   Proof of Proposition 17

**Proposition 17.** *Suppose $\{X_t,\mathcal{F}_t\}$ is a MDS such that $\mathbb{E}[X_t^2|\mathcal{F}_t]\leq\sigma_t^2$ and $|X_t|\leq b_t$ almost everywhere for all $t$ for some sequence of random variable $\{\sigma_t,b_t\}$ such that $\sigma_t,b_t$ is $\mathcal{F}_{t-1}$-measurable. Then $X_t$ is $(\sigma_t,2b_t)$ sub-exponential.*

*Proof.* Suppose $\lambda\leq 1/2b_t$. Then we compute for any $k\geq 2$:

$$\mathbb{E}\left[\frac{\lambda^k X_t^k}{k!}\mid\mathcal{F}_{t-1}\right]\leq\frac{\lambda^k b_t^{k-2}}{k!}\mathbb{E}[X_t^2|\mathcal{F}_{t-1}]$$
$$\leq\frac{\lambda^k b_t^{k-2}\sigma_t^2}{k!}$$
$$\leq\frac{\lambda^2\sigma_t^2}{2^{k-2}k!}$$

Further, since $X_t$ is a MDS, we also have $\mathbb{E}\left[\lambda X_t\mid\mathcal{F}_{t-1}\right]=0$. Therefore:

$$\mathbb{E}[\exp(\lambda X_t)|\mathcal{F}_{t-1}]\leq 1+\sum_{k=2}^{\infty}\frac{\lambda^2\sigma_t^2}{2^{k-2}k!}$$
$$\leq 1+\frac{\lambda^2\sigma_t^2}{2}\sum_{i=0}^{\infty}\frac{1}{2^i}$$
$$= 1+\frac{\lambda^2\sigma_t^2}{2}$$
$$\leq\exp(\lambda^2\sigma_t^2/2)$$

where the last line uses the identity $1+x\leq\exp(x)$. $\square$

### E.5 Classical Concentration Bound for MDS

**Lemma 22** (Scaled Sub-exponential). *Suppose that $\{X_t, \mathcal{F}_t\}$ is a MDS such that $\mathbb{E}[X_t|\mathcal{F}_t] = 0$, $\mathbb{E}[X_t^2 \mid \mathcal{F}_t] \leq \sigma$ and $|X_t| \leq b$ almost surely for some fixed $\sigma, b$. Let $\nu$ be an arbitrary fixed number, then for all $t$ with probability at least $1 - \delta$:*

$$\nu X_t \leq 2|\nu|b\log\frac{1}{\delta} + |\nu|\sigma\sqrt{2\log\frac{1}{\delta}}$$

*Proof.* First we have $\mathbb{E}[\nu^2 X_t^2] \leq \nu^2\sigma^2, |\nu X_t| \leq |\nu|b$ almost surely and $\{\nu X_t, \mathcal{F}_t\}$ is also a MDS. By Proposition 17, $\nu X_t$ is $(|\nu|\sigma, 2|\nu|b)$ sub-exponential. Use definition 1 and tower rule,

$$\mathbb{E}[\exp(\lambda\nu X_t)] = \mathbb{E}[\mathbb{E}[\exp(\lambda\nu X_t) \mid \mathcal{F}_{t-1}]] = \exp(\lambda^2|\nu|\sigma^2/2)$$

for $\lambda \leq 1/(2|\nu|b)$. The above inequality make us returns to the standard result of independent sub-exponential random variable.

$$\mathbb{E}[\exp(\lambda\nu X_t)] = \exp(\lambda^2|\nu|\sigma^2/2)$$

Hence, from the standard sub-exponential tails

$$P[\nu X_t \geq a] \leq \begin{cases} \exp(-a^2/2\nu^2\sigma^2) & 0 \leq a \leq \sigma^2|\nu|/2b, \\ \exp(-a/2|\nu|b) & a > \sigma^2|\nu|/2b \end{cases}$$

Set above quantities as $\delta$ and rearrange for $a$:

$$a = \max\left(2|\nu|b\log\frac{1}{\delta}, \sqrt{2|\nu|^2\sigma^2\log\frac{1}{\delta}}\right)$$

Hence with probability at least $1 - \delta$

$$\nu X_t \leq \max\left(2|\nu|b\log\frac{1}{\delta}, \sqrt{2|\nu|^2\sigma^2\log\frac{1}{\delta}}\right)$$

$$\leq 2|\nu|b\log\frac{1}{\delta} + |\nu|\sigma\sqrt{2\log\frac{1}{\delta}}$$

$\square$

**Lemma 23** (Scaled Sub-exponential Sum). *Suppose that $\{X_t, \mathcal{F}_t\}$ is a MDS such that $\mathbb{E}[X_t \mid \mathcal{F}_t] = 0$, $\mathbb{E}[X_t^2 \mid \mathcal{F}_t] \leq \sigma$ and $|X_t| \leq b$ almost surely. Let $\nu$ be an arbitrary fixed number, then for all $t$ with probability at least $1 - \delta$:*

$$\sum_{i=1}^{t}\nu X_i \leq 2|\nu|b\log\frac{1}{\delta} + |\nu|\sigma\sqrt{2t\log\frac{1}{\delta}}, \quad \forall t$$

*Proof.*

$$\mathbb{E}[e^{\lambda\sum_{i=1}^{t}\nu X_i}] = \prod_{i=1}^{t}\mathbb{E}[e^{\lambda\nu X_i}]$$

for $|\lambda| \leq \frac{1}{2|\nu|b}$, invoke Lemma 22

$$\leq \exp(t\lambda^2|\nu|\sigma^2/2)$$

Let $\sigma' = \sigma\sqrt{t}, \sigma' \in \mathbb{R}$, hence we can directly use the concentration bound derived in Lemma 22 to complete the proof. $\square$

**Lemma 24** (Sub-exponential Squared). *Suppose $\{X_t, \mathcal{F}_t\}$ is a MDS with $|X_t| \leq b$ and $\mathbb{E}[X_t^2 \mid \mathcal{F}_t] \leq \sigma^2$ almost surely for some fixed $\sigma, b$. Then with probability at least $1 - \delta$*

$$\sum_{t=1}^{T}X_t^2 \leq \frac{3\sigma^2}{2}T + \frac{5}{3}b^2\log\frac{1}{\delta}$$

*Proof.* Let $Z_t = X_t^2 - \mathbb{E}[X_t^2]$, $Z_0, \cdots, Z_T$ is a martingale difference sequence adapted to $\mathcal{F}_t$. Also $|Z_t| < X_t^2 \leq b^2$. Also

$$\mathbb{E}[Z_t^2] = \mathbb{E}[|Z_t| \cdot |Z_t|] \leq b^2 \mathbb{E}[|Z_t|] \leq b^2 \mathbb{E}[x_t^2] \leq b^2 \sigma^2$$

From Freedman's inequality for martingale sequences (see e.g. Tropp [2011], or Lemma 11 in Cutkosky and Mehta [2021] for the form we use here), with probability at least $1 - \delta$,

$$\sum_{t=1}^{T} Z_t \leq \frac{2}{3} b^2 \log \frac{1}{\delta} + \sigma b \sqrt{2T \log \frac{1}{\delta}}$$

Rearranging the definition of $Z_t$, with probability at least $1 - \delta$:

$$\sum_{t=1}^{T} X_t^2 = \sum_{t=1}^{T} Z_t + \sum_{t=1}^{T} \mathbb{E}[X_t^2]$$

$$\leq \frac{2}{3} b^2 \log \frac{1}{\delta} + \sigma b \sqrt{2T \log \frac{1}{\delta}} + T\sigma^2 \tag{26}$$

by young's inequality $\sigma b \leq \sigma^2/(2\lambda) + \lambda b^2/2$, set $\lambda = \sqrt{2 \log \frac{1}{\delta}/T}$, we complete the proof $\qquad \square$

### E.6 Another Technical Lemma

**Lemma 25.** *Suppose $Z$ is such that*

$$Z \leq A \log \left( B \left[ \log \left( \left[ C\sqrt{Z} \right]_1 \right) + 2 \right]^2 \right)$$

*for some constants $A, B, C \geq 0$, where $[x]_1 = \max(1, x)$ Then*

$$Z \leq 4A \log \left( 4\sqrt{B} \log(e + 16C^2 A) \right)$$

*Proof.* Expanding the logarithms in the given bound on $Z$, we have:

$$Z \leq A \log(B) + 2A \log(\log([C\sqrt{Z}]_1) + 2)$$

Now, since $a + b \leq \max(2a, 2b)$, we have that either $Z \leq 2A \log(B)$, or

$$Z \leq 4A \log(\log([C\sqrt{Z}]_1) + 2)$$

In the first case, we are done since $2A \log(B) \leq 4A \log \left( 4\sqrt{B} \log(e + 16C^2 A) \right)$, so let us consider only the second case $Z \leq 4A \log(\log([C\sqrt{Z}]_1) + 2)$. Now, we define the function

$$f(x) = 4A \log(\log([C\sqrt{x}]_1)) + 2) \tag{27}$$

Notice that for all $x \geq A$, we have either $f'(x) = 0$ or $C\sqrt{x} \geq 1$ and

$$f'(x) = \frac{2A}{x \log(C\sqrt{x}) + 2x} < 1$$

so that if $Z_\star$ is any value satisfying $Z_\star \geq A$ and $Z_\star \geq f(Z_\star)$, then we must have $Z \leq Z_\star$: otherwise $f(Z) = f(Z_\star) + \int_{Z_\star}^{Z} f'(z)dz < f(Z_\star) + Z - Z_\star \leq Z$, a contradiction. Let us consider:

$$Z_\star = 4A \log \left[ 4 \log(e + QA) \right]$$

for some to-be-specified $Q \geq 0$. Notice that this $Z_\star$ clearly satisfies $Z_\star \geq 8A \log(2)$. Let us show that $Z_\star \geq f(Z_\star)$.

Again using $x + y \leq 2 \max(x, y)$, we have:

$$f(Z_\star) \leq 4A \log \left( \max(4, \ 2 \max(C\sqrt{Z_\star}, 1)) \right)$$

$$= \max\left(4A\log(4),\ 4A\log(2\log(\max(C\sqrt{Z_\star}, 1)))\right)$$

Now, if $f(Z_\star) \leq 4A\log(4)$ then we clearly have $f(Z_\star) \leq Z_\star$ as desired. So, let us focus on the case $f(Z_\star) \leq 4A\log(2\log(\max(C\sqrt{Z_\star}, 1)))$. Next, if $C\sqrt{Z_\star} \leq e$, then we have $f(Z_\star) \leq 4A\log(2) \leq Z_\star$ again as desired. Thus we may further restrict to the case $C\sqrt{Z_\star} \geq e$ so that $\max(C\sqrt{Z_\star}, 1) = C\sqrt{Z_\star}$ and the bound on $f(Z_\star)$ is $4A\log(2\log(C\sqrt{Z_\star}))$. Plugging in our expression for $Z_\star$:

$$f(Z_\star) \leq 4A\log\left[2\log\left[2C\sqrt{A\log\left[4\log(e + QA)\right]}\right]\right]$$

Comparing with the expression for $Z_\star$, we see that to establish $Z_\star \geq f(Z_\star)$, it suffices to show:

$$(e + QA)^2 \geq 2C\sqrt{A\log\left[4\log(e + QA)\right]}$$

Now, using $\log(x) \leq x$ twice:

$$2C\sqrt{A\log(4\log(e + QA))} \leq 4C\sqrt{A}\sqrt{(e + QA)} \tag{28}$$
$$= \sqrt{16C^2 A}\sqrt{e + QA} \tag{29}$$
$$\leq \sqrt{e + 16C^2 A}\sqrt{e + QA} \tag{30}$$

if we set $Q = 16C^2$, we will have:

$$= e + QA \tag{31}$$
$$\leq (e + QA)^2 \tag{32}$$

Thus, by setting $Q = 16C^2$, we will have $Z_\star \geq f(Z_\star)$ and so we have $f(Z_\star) \leq Z_\star$, which implies

$$Z \leq Z_\star$$
$$= 4A\log\left[4\log(e + 16C^2 A)\right]$$
$$\leq 4A\log\left(4\sqrt{B}\log(e + 16C^2 A)\right)$$

as desired since $B \geq 1$. $\qquad\square$