# OpenReview forum: "Parameter-free Regret in High Probability with Heavy Tails"
_NeurIPS.cc/2022/Conference — NeurIPS 2022 Accept_

### Official Review · Reviewer_Wska · 2022-07-10

**Rating:** 4
**Confidence:** 4
**Soundness:** 3 good
**Presentation:** 3 good
**Contribution:** 3 good

**Summary:**

The paper studies online convex optimization over unbounded domains and aims to provide a high-probability regret bound given access to only potentially heavy-tailed gradient information. Towards this end, the authors propose several interesting ideas to overcome the anti-concentration issue due to the heavy-tailed noise. The basic idea is to add some regularization to cancel additional term introduced by the heavy-tailed gradient. The results are new and the techniques are interesting.

**Questions:**

no

**Ethics Review Area:**

["I don’t know"]

**Limitations:**

see comments in Strengths And Weaknesses

**Strengths And Weaknesses:**

The paper is generally of good quality, including the writing and presentation logic. The results for unbounded online convex optimization are new, to the best of my knowledge. I have two concerns as stated below.

1) It is acknowledged that heavy-tailed issues and parameter-free bounds are interesting to the community. But does it have practical appeals when combining the two? The authors are encouraged to motivate their paper with a concrete and practical example in real applications, such that the readers will be more convinced to believe that we indeed need to pay some effort to study this problem. The current paper presentation does not consider this, which makes the reader think of "creating" this problem to solve.

2) The solution for this problem is kind of simple. I am not saying that simplicity is a drawback, but some discussions on whether the idea can be applied in broader applications/theoretical problems would strengthen the technical contribution. Additionally, the paper lacks discussions on the computational complexity -- for example, step 5 in Algorithm 1 is like a fixed-point equation, how to solve it in an efficient way? what's the overall computational complexity? All those stuff should be discussed in the paper.

---

> ### Author Response · Authors · 2022-08-01
> **Response to Reviewer Wska for Paper12527**
>
> Thanks for your helpful comments and questions, we hope our answers below will help clarify the contribution. We think that incorporating these clarifications will improve our manuscript. If you agree, we hope you will consider increasing your score.
>
> First, we’d like to clarify the extent of our contribution: It is *not* limited to the setting of heavy-tailed noise, which was perhaps emphasized too much in our title. Our contribution is better characterized as a comprehensive suite of techniques for *high-probability* analysis of parameter-free methods. This includes the first high probability bounds *of any kind*, including for the much simpler setting of *bounded* gradient estimates. The prior state of the art usually encompassed only in-expectation bounds for bounded gradients, or at best in-expectation bounds for sub-exponential gradients. Our results go significantly further: not only do we obtain high-probability bounds, our techniques are powerful enough to move beyond bounded gradients to not just sub-exponential but also general heavy-tailed gradients. We emphasize that *even if we had only obtained a high-probability bound for bounded gradients, this would still be a meaningful improvement on the state of the art*.
>
> Next, we agree that we were remiss in motivating the interest in these high probability bounds. We will add some discussion to the paper to make it clear why this is an important problem. Intuitively, a high probability bound means that we can be confident that any given run of an algorithm will perform very well. This is crucially important in the online setting in which we must make irrevocable decisions. It is also important in the standard stochastic optimization setting encountered throughout machine learning as it  ensures that even a single potentially very expensive training run will produce a good result. This goal naturally *synergizes* with the overall objective of adaptive and parameter-free algorithms: both attempt to ensure high-quality performance after a single pass over the data.
>
> Now, heavy tailed gradient estimates specifically have been an ongoing area of interest in the community. In addition to inherent theoretical appeal, these actually form an emerging focus in practice as empirical results suggest that the gradients of some large neural network architectures display heavy tails [1-3]. We are certainly not the first to study these topics in optimization. See [4-6] for recent papers and further motivation.
>
> In terms of computational complexity for the fixed point equation (line 5 in algorithm 1). It can be solved using bisection methods in $O(log \frac{1}{\epsilon_0})$ to any desired precision $\epsilon_0$. We will add a clarification to the paper about this, thank you for bringing it up! As a result, our algorithm for $d$-dimensional heavy-tailed gradients takes $O(d+log \frac{1}{\epsilon_0})$ operations for each step $t$, which is the same as online gradient descent as long as $\epsilon_0 \ge 2^{-d}$. Please see the response for reviewer WCjH for more details regarding complexity.
>
> [1] Zhou, Pan, et al. "Towards theoretically understanding why sgd generalizes better than adam in deep learning." Advances in Neural Information Processing Systems 33 (2020): 21285-21296.
>
> [2] Simsekli, Umut, Levent Sagun, and Mert Gurbuzbalaban. "A tail-index analysis of stochastic gradient noise in deep neural networks." International Conference on Machine Learning. PMLR, 2019.
>
> [3] Zhang, Jingzhao, et al. "Why adam beats sgd for attention models." (2019).
>
> [4] Li, Shaojie, and Yong Liu. "High Probability Guarantees for Nonconvex Stochastic Gradient Descent with Heavy Tails." International Conference on Machine Learning. PMLR, 2022.
>
> [5] Gurbuzbalaban, Mert, Umut Simsekli, and Lingjiong Zhu. "The heavy-tail phenomenon in SGD." International Conference on Machine Learning. PMLR, 2021.
>
> [6] Madden, Liam, Emiliano Dall'Anese, and Stephen Becker. "High probability convergence bounds for stochastic gradient descent assuming the polyak-lojasiewicz inequality." arXiv preprint arXiv:2006.05610 (2020).

---

### Official Review · Reviewer_tcgd · 2022-07-10

**Rating:** 5
**Confidence:** 3
**Soundness:** 2 fair
**Presentation:** 3 good
**Contribution:** 2 fair

**Summary:**

This paper proposes new algorithms for online convex optimization over unbounded domains. The authors provide a parameter-free regret in high probability with heavy tails. First, they introduce a parameter-free algorithm for sub-exponential $g_t$ that achieves regret $O(\epsilon+|u|\sqrt{T})$ in high probability. Then, they extend to heavy-tailed $g_t$ by employing clipping on the heavy-tail. Last, they extend the algorithms to arbitrary dimensions.



**Questions:**

 Is this possible to set a appropriate $\tau$ even if we did not have any information about $\sigma$?

**Limitations:**

 1. Same as weakness #3, $\tau$ may not be known, which could be a challenge when we run the algorithm 2.
 2. Two regularization functions introduce suboptimal logarithmic factors: The first one introduces a higher logarithmic dependence on T, and the second introduces a higher logarithmic dependence on |u|.

**Strengths And Weaknesses:**

Strengths:
1. The authors present a framework for building parameter-free algorithms that achieve high probability regret bounds for heavy-tailed subgradient estimates: high probability bounds were previously unavailable even for the restricted setting of bounded subgradient estimates.
2. They use Huber loss to construct a regularizer to ensure that it achieves low regret with a high probability in Section 4, and define another regularization function (from clipped $\hat{g}_t$) for the purpose of bias cancellation in Section 5.
3. They also extend the work to arbitrary dimensions.

Weaknesses:
1. The paper needs proof reading: For example: In page 2 contribution and Organization: We introduce a parameter-free algorithm for supexponential->sub-exponential.... ;  "Secondly, we extend to heavy tailed "..-> heavy-tailed
2. In Section 4,   $|g_t|\leq b$ is mentioned. Note that, this b is $O(\sqrt{T})$. It is better to use $b_t$ because $b$ is used for constant value.
3. In theorem 4, $\tau$ may be unknown because it needs the knowledge of $\sigma$. Algorithm 2 could not achieve Thm 4's regret bound if we do not know $\tau$.

---

> ### Author Response · Authors · 2022-08-01
> **Response to Reviewer tcgd for Paper12527**
>
> Thank you for your valuable comments on our submission, they will be very helpful in improving the paper! Below, we provide some response to your specific questions:
>
> For weakness 2: We intentionally used the constant $b$ when writing the condition $\|g_t\|\le b$ to consider a fixed sub-exponential parameter. Notice that the the statements of Theorems 2 and 3 are consistent with this notation. The confusion may arise from the use of varying $b_t$ in Definition 1. We allow for this greater generality that is not used in the main text. Because in our proof of the martingale analysis (appendix), we need to control the martingale difference sequence $\langle \nabla \ell_t(w_t) - g_t, w_t\rangle$. This sequence does have a time-varying sub-exponential parameter $b_t = b \|w_t\|$ and controlling this properly is a key technical component of our analysis. So, in summary, while a time-varying sub-exponential parameter $b_t$ is important in our analysis, it is not present in the main theorem statements. We acknowledge the use of $b$ vs $b_t$ in these subtly different contexts is a bit confusing and we will revise the manuscript to clarify the notation.
>
> For weakness 3: You’re right, it would be better to not require knowledge of $\sigma$. However, adapting to unknown variance in the heavy tailed setting is a difficult problem of significant interest in its own right beyond the online learning scenario. For example, typical approaches for simple fundamental problems like mean estimation with heavy tails usually require information about similar variance-like parameters of the distribution, see example [1] or the excellent survey [2]. In our case, achieving a meaningful regret bound even with this knowledge is already an accomplishment. Beyond the references presented in our submission, other recent studies on optimization algorithms with heavy tails that suffer similar restrictions include [3-5]. Nevertheless, we agree that the ideal algorithm would not have this limitation, and we will make sure to include a discussion of this in the final copy.
>
>
> [1] Tsai, Che-Ping, et al. "Heavy-tailed streaming statistical estimation." International Conference on Artificial Intelligence and Statistics. PMLR, 2022.
>
> [2] Lugosi, Gábor, and Shahar Mendelson. "Mean estimation and regression under heavy-tailed distributions: A survey." Foundations of Computational Mathematics 19.5 (2019): 1145-1190.
>
> [3] Li, Shaojie, and Yong Liu. "High Probability Guarantees for Nonconvex Stochastic Gradient Descent with Heavy Tails." International Conference on Machine Learning. PMLR, 2022.
>
> [4] Kamath, Gautam, Xingtu Liu, and Huanyu Zhang. "Improved rates for differentially private stochastic convex optimization with heavy-tailed data." International Conference on Machine Learning. PMLR, 2022.
>
> [5] Lee, Kyungjae, et al. "Optimal algorithms for stochastic multi-armed bandits with heavy tailed rewards." Advances in Neural Information Processing Systems 33 (2020): 8452-8462.

---

### Official Review · Reviewer_WCJh · 2022-07-11

**Rating:** 7
**Confidence:** 3
**Soundness:** 3 good
**Presentation:** 3 good
**Contribution:** 3 good

**Summary:**

The authors consider online convex optimization over unbounded domains with potentially heavy-tailed subgradient estimates as feedback. Precisely they assume that the p-th(\in(1,2]) moment of the subgradient is bounded. They propose an algorithm that with high probability has regret against any fixed comparator u of order O(|u|\sqrt{T}) where T is the horizon. Furthermore, the algorithm is parameter-free in the sense that it does not need to know the norm |u|. This algorithm relies on two novel regularizers that share similarities with the Huber loss. One to deal with the fact that the domain is unbounded and the other to handle the heavy-tailed subgradient.


**Questions:**

Specific comments:

-L30: Since at since point the role of \epsilon is not clear why not hide it in the \tilde{O} notation?

-L81: log factors with respect to which quantities of the problem, i.e. not log factor of exp(T) ?

-L128: G is not defined?

-L199: At this point it is not completely clear why you need two algorithms \cA_1 and \cA_2.

-L201: Maybe you should give at least one of these algorithm in then appendix.

-L208: Does the result holds with probability one?

-L266: g_1

-L281: Algorithm 3, Require, g_t instead of \hat{g}_t?

-L284: Can you discuss how the obtained bound mathc or not the lower bound you mention in the introduction? Can you also provide the overall time/space complexity of Algorithm 4?

-L427: Theorem 1 and a dot is missing.

-L428: R instead of \mathbb{R}

-L429: parenthesis and the statement of the lemma is not very clear.

-L432: punctuation of the equations is missing.

-L435: * not at the right place.

-L444: What is \bar{G}?

-L459: Can you detail how you applied Lemma 7 (choice of G and \bar{G}?

-L462: inf over y^\star is missing.

-L466, below: it seems that the term |A|^2uH[...] is repeated.

-L470: The first sentence is not clear.

-L483: The capital for Theorem, Section, Algorithm, Appendix is a bit inconsistent.

-L542: w_t^2\epsilon_t^2

**Limitations:**

Yes.

**Strengths And Weaknesses:**

Contributions:
-parameter-free algorithm for (1D) supexponential stochastic subgradient estimate, novelty: medium, relevance: high.
-parameter-free algorithm for heavy-tailed stochastic subgradient estimate, novelty: medium, relevance: high.

The paper is well-written, and the intuition, in particular, is clearly described. Maybe a small description of the role of \cA_1 and \cA_2 in Algorithm 1 could make the presentation clearer. The proofs seem correct. I did not read the appendix C, D, E. Note that the writing in the appendix could be improved, see specific comments. The considered setting and the obtained results may look a bit incremental but I think that the new regularizers and the associated analyses developed to overcome the technical difficulties are a valuable contribution.

---

> ### Author Response · Authors · 2022-08-01
> **Response to Reviewer WCJh for Paper12527**
>
> Thanks for the detailed and insightful comments! We list below responses to some specific questions and will address all of them to improve the final copy.
>
> -L81: Our $\tilde{O}(\cdot)$ hides constants and polylogarithmic factors in $T$, $\|u\|$, $G$ and $\epsilon$.
>
> -L199: We agree they seem somewhat mysterious at first. First, recall $x_t, y_t$ are outputs of $\mathcal{A}_1$ and $\mathcal{A}_2$, respectively. And $w_t$ is the output of Algorithm 1. Notice the $w_t = x_t - y_t \nabla \psi(w_t)$ in Algorithm 1 (line 5). Intuitively, $\mathcal{A}_1$ is providing some kind of “initial” guess for what the final output $w_t$ should be by simply playing an online learning game using the gradients $g_t$. $\mathcal{A}_2$ is responsible for “correcting” this $x_t$ by learning the amount of regularization to apply (notice that $w_t = x_t - y_t\nabla \psi(w_t)$ is an implicit gradient descent step from $x_t$ to $w_t$ using the objective $y_t \psi$, where $y_t$ is learned by $\mathcal{A}_2$, $y_t$ is analogue to step size). We will add more detail to the text to clarify this notion.
>
> -L208: Theorem 2 is a deterministic guarantee - recall that we defined $R_T^A=\sum_{t=1}^T \hat \ell_t(w_t)-\hat\ell_t(u)$ with $\hat \ell_t(w) = \langle g_t, w\rangle + \psi(w)$, so that a deterministic guarantee is possible. We will clarify this in the text.
>
> -L284: In Theorem 5, the polynomial dependence in $T$ is $T^{1/p}$, so it matches with the lower bound mentioned in the introduction up to logarithmic factors.
> The overall space complexity of Algorithm 4 is $O(d)$. The time-complexity involves solving the implicit (1D) equation for $w_t$ in line 5 of Algorithm 1, which can be done by binary search: we can show that $y_t\nabla \psi_t(w_t) + w_t$ is monotonic in $w_t$, and so we can solve for $w_t$ to any desired precision $\epsilon_0$ in $\log(1/\epsilon_0)$ time. The rest of the computations takes $O(d)$ time for a total of $O(d + \log(1/\epsilon_0))$, which is $O(d)$ as long as $\epsilon \ge 2^{-d}$ (i.e. in practice as soon as $d\ge 64$). This is the same complexity as gradient descent. Thanks for pointing out this missing part of the story - we will of course update the manuscript to include these calculations.
>
> -L459: In many prior works on parameter-free algorithms, it is assumed that the Lipschitz constant is 1. We do not wish to make this restriction, and indeed in a few places we need to instantiate instances of parameter-free algorithms with different Lipschitz constants. However, in these same places, we *also* need to control $\text{Regret}(0)$. Lemma 7 provides a general way to do this: starting from any parameter-free algorithm with a given assumed Lipschitz bound of $G$ and $\text{Regret}(0)\le \epsilon$, we can convert it into one with an assumed bound of $\bar{G}$ and $\text{Regret}(0)\le \bar{\epsilon}$ for any desired $\bar{G}$ and $\bar{\epsilon}$. The key point of this Lemma is that appropriate transformation of standard parameter-free algorithms will yield algorithms that work well with arbitrary Lipschitz constants.
> So, for example, usually $G=1$ (as prior work often assumes $G=1$). In the proof of Theorem 2, Lemma 7 was used twice, once with  $\bar{G}=b+H$), and once with $\bar{G}=H(b+H)$. These are simply bounds on the gradients provided to the base algorithms in Algorithm 1. The value of $\bar{\epsilon}$ is just $\epsilon$ in our case. We have updated all the proofs in the appendix to include these details when Lemma 7 is applied.

---

> > ### Comment · Reviewer_WCJh · 2022-08-08
> > **Post-rebuttal**
> >
> > Thanks for the response. I read the other reviews and the author's rebuttal. I'll keep my initial evaluation.

---

### Author Response · Authors · 2022-08-01
**Response to All Reviewers**

We thank all the reviewers for the efforts and useful feedback. We provided an updated version of the manuscript with alterations that address the more technical questions (highlighted in blue) and Lemma 6. We will use the extra page for the final copy to address the more expository comments. For specific questions from reviewers, we will respond to each of you below.

---

### Meta-Review · Area_Chair_9Pf5 · 2022-08-22

**Recommendation:** Accept
**Confidence:** Certain

**Metareview:**

The paper makes an interesting contribution to the literature on online convex optimization with heavy-tailed stochastic gradient including infinite variance. While two reviewers (tcgd and WCJh) were positive about the paper, one reviewer (Wska) had concerns about the motivation and the technical contributions (commenting about it at a very-high level).

In my own reading of the paper, I find it is well-written, highlighting the main challenges and the proof strategy they use to overcome the challenges. The regret bounds obtained are also optimal in a certain sense as they achieve lower bounds obtained recently.

Hence it is recommended for acceptance at Neurips.

**Award:**

No

---

### Decision · Program_Chairs · 2022-09-14

Accept